# POPULATION-GUIDED PARALLEL POLICY SEARCH FOR REINFORCEMENT LEARNING

**Whiyoung Jung, Giseung Park, Youngchul Sung**[*]
School of Electrical Engineering
Korea Advanced Institute of Science and Technology
{wy.jung, gs.park, ycsung}@kaist.ac.kr

## ABSTRACT

In this paper, a new population-guided parallel learning scheme is proposed to enhance the performance of off-policy reinforcement learning (RL). In the proposed scheme, multiple identical learners with their own value-functions and policies share a common experience replay buffer, and search a good policy in collaboration with the guidance of the best policy information. The key point is that the information of the best policy is fused in a soft manner by constructing an augmented loss function for policy update to enlarge the overall search region by the multiple learners. The guidance by the previous best policy and the enlarged range enable faster and better policy search. Monotone improvement of the expected cumulative return by the proposed scheme is proved theoretically. Working algorithms are constructed by applying the proposed scheme to the twin delayed deep deterministic (TD3) policy gradient algorithm. Numerical results show that the constructed algorithm outperforms most of the current state-of-the-art RL algorithms, and the gain is significant in the case of sparse reward environment.

## 1 INTRODUCTION

RL is an active research field and has been applied successfully to games, simulations, and actual environments. With the success of RL in relatively easy tasks, more challenging tasks such as sparse reward environments (Oh et al. (2018); Zheng et al. (2018); Burda et al. (2019)) are emerging, and developing good RL algorithms for such challenging tasks is of great importance from both theoretical and practical perspectives. In this paper, we consider parallel learning, which is an important line of RL research to enhance the learning performance by having multiple learners for the same environment. Parallelism in learning has been investigated widely in distributed RL (Nair et al. (2015); Mnih et al. (2016); Horgan et al. (2018); Barth-Maron et al. (2018); Espeholt et al. (2018)), evolutionary algorithms (Salimans et al. (2017); Choromanski et al. (2018); Khadka & Tumer (2018); Pourchot & Sigaud (2019)), concurrent RL (Silver et al. (2013); Guo & Brunskill (2015); Dimakopoulou & Van Roy (2018); Dimakopoulou et al. (2018)) and population-based training (PBT) (Jaderberg et al. (2017; 2018); Conti et al. (2018)). In this paper, in order to enhance the learning performance, we apply parallelism to RL based on a population of policies, but the usage is different from the previous methods.

One of the advantages of using a population is the capability to evaluate policies in the population. Once all policies in the population are evaluated, we can use information of the best policy to enhance the performance. One simple way to exploit the best policy information is that we reset the policy parameter of each learner with that of the best learner at the beginning of the next $M$ time steps; make each learner perform learning from this initial point for the next $M$ time steps; select the best learner again at the end of the next $M$ time steps; and repeat this procedure every $M$ time steps in a similar way that PBT does (Jaderberg et al. (2017)). We will refer to this method as the resetting method in this paper. However, this resetting method has the problem that the search area covered by all $N$ policies in the population collapses to one point at the time of parameter copying and thus the search area can be narrow around the previous best policy point. To overcome such disadvantage, instead of resetting the policy parameter with the best policy parameter periodically,

---

[*]Corresponding author

we propose using the best policy information in a soft manner. In the proposed scheme, the shared best policy information is used only to guide other learners' policies for searching a better policy. The chief periodically determines the best policy among all learners and distributes the best policy parameter to all learners so that the learners search for better policies with the guidance of the previous best policy. The chief also enforces that the $N$ policies are spread in the policy space with a given distance from the previous best policy point so that the search area by all $N$ learners maintains a wide area and does not collapse into a narrow region.

The proposed Population-guided Parallel Policy Search (P3S) learning method can be applied to any off-policy RL algorithms and implementation is easy. Furthermore, monotone improvement of the expected cumulative return by the P3S scheme is theoretically proved. We apply our P3S scheme to the TD3 algorithm, which is a state-of-the-art off-policy algorithm, as our base algorithm. Numerical result shows that the P3S-TD3 algorithm outperforms the baseline algorithms both in the speed of convergence and in the final steady-state performance.

## 2 BACKGROUND AND RELATED WORKS

**Distributed RL** Distributed RL is an efficient way of taking advantage of parallelism to achieve fast training for large complex tasks (Nair et al. (2015)). Most of the works in distributed RL assume a common structure composed of multiple actors interacting with multiple copies of the same environment and a central system which stores and optimizes the common Q-function parameter or the policy parameter shared by all actors. The focus of distributed RL is to optimize the Q-function parameter or the policy parameter fast by generating more samples for the same wall clock time with multiple actors. For this goal, researchers investigated various techniques for distributed RL, e.g., asynchronous update of parameters (Mnih et al. (2016); Babaeizadeh et al. (2017)), sharing an experience replay buffer (Horgan et al. (2018)), GPU-based parallel computation (Babaeizadeh et al. (2017); Clemente et al. (2017)), GPU-based simulation (Liang et al. (2018)) and V-trace in case of on-policy algorithms (Espeholt et al. (2018)). Distributed RL yields performance improvement in terms of the wall clock time but it does not consider the possible enhancement by interaction among a population of policies of all learners like in PBT or our P3S. The proposed P3S uses a similar structure to that in (Nair et al. (2015); Espeholt et al. (2018)): that is, P3S is composed of multiple learners and a chief. The difference is that each learner in P3S has its own Q or value function parameter and policy parameter, and optimizes the parameters in parallel to search in the policy space.

**Population-Based Training** Parallelism is also exploited in finding optimal parameters and hyper-parameters of training algorithms in PBT (Jaderberg et al. (2017; 2018); Conti et al. (2018)). PBT trains neural networks, using a population with different parameters and hyper-parameters in parallel at multiple learners. During the training, in order to take advantage of the population, it evaluates the performance of networks with parameters and hyper-parameters in the population periodically. Then, PBT selects the best hyper-parameters, distributes the best hyper-parameters and the corresponding parameters to other learners, and continues the training of neural networks. Recently, PBT is applied to competitive multi-agent RL (Jaderberg et al. (2018)) and novelty search algorithms (Conti et al. (2018)). The proposed P3S uses a population to search a better policy by exploiting the best policy information similarly to PBT, but the way of using the best policy information is different. In P3S, the parameter of the best learner is not copied but used in a soft manner to guide the population for better search in the policy space.

**Guided Policy Search** Our P3S method is also related to guided policy search (Levine & Koltun (2013); Levine et al. (2016); Teh et al. (2017); Ghosh et al. (2018)). Teh et al. (2017) proposed a guided policy search method for joint training of multiple tasks in which a common policy is used to guide local policies and the common policy is distilled from the local policies. Here, the local policies' parameters are updated to maximize the performance and minimize the KL divergence between the local policies and the common distilled policy. The proposed P3S is related to guided policy search in the sense that multiple policies are guided by a common policy. However, the difference is that the goal of P3S is not learning multiple tasks but learning optimal parameter for a common task as in PBT. Hence, the guiding policy is not distilled from multiple local policies but chosen as the best performing policy among multiple learners.

**Exploiting Best Information** Exploiting best information has been considered in the previous works (White & Sofge (1992); Oh et al. (2018); Gangwani et al. (2019)). In particular, Oh et al. (2018); Gangwani et al. (2019) exploited past good experiences to obtain a better policy, whereas P3S exploits the current good policy among multiple policies to obtain a better policy.

## 3 POPULATION-GUIDED PARALLEL POLICY SEARCH

The overall structure of the proposed P3S scheme is described in Fig. 1. We have $N$ identical parallel learners with a shared common experience replay buffer $\mathcal{D}$, and all $N$ identical learners employ a common base algorithm which can be any off-policy RL algorithm. The execution is in parallel. The $i$-th learner has its own environment $\mathcal{E}^i$, which is a copy of the common environment $\mathcal{E}$, and has its own value function (e.g., Q-function) parameter $\theta^i$ and policy parameter $\phi^i$. The $i$-th learner interacts with the environment copy $\mathcal{E}^i$ with additional interaction with the chief, as shown in

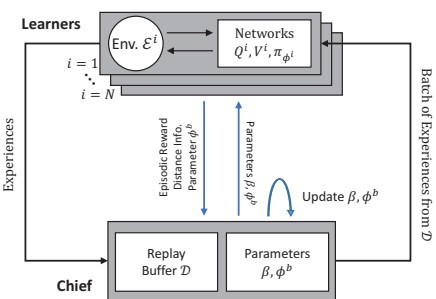

Figure 1: The overall structure of P3S

Fig. 1. At each time step, the $i$-th learner performs an action $a_t^i$ to its environment copy $\mathcal{E}^i$ by using its own policy $\pi_{\phi^i}$, stores its experience $(s_t^i, a_t^i, r_t^i, s_{t+1}^i)$ to the shared common replay buffer $\mathcal{D}$ for all $i = 1, 2, \cdots, N$. Then, each learner updates its value function parameter and policy parameter once by drawing a mini-batch of size $B$ from the shared common replay buffer $\mathcal{D}$ by minimizing its own value loss function and policy loss function, respectively.

Due to parallel update of parameters, the policies of all learners compose a population of $N$ different policies. In order to take advantage of this population, we exploit the policy information from the best learner periodically during the training like in PBT (Jaderberg et al. (2017)). Suppose that the Q-function parameter and policy parameter of each learner are initialized and learning is performed as described above for $M$ time steps. At the end of the $M$ time steps, we determine who is the best learner based on the average of the most recent $E_r$ episodic rewards for each learner. Let the index of the best learner be $b$. Then, the policy parameter information $\phi^b$ of the best learner can be used to enhance the learning of other learners for the next $M$ time steps. Instead of copying $\phi^b$ to other learners like in PBT, we propose using the information $\phi^b$ in a soft manner. That is, during the next $M$ time steps, while we set the loss function $\widetilde{L}(\theta^i)$ for the Q-function to be the same as the loss $L(\theta^i)$ of the base algorithm, we set the loss function $\widetilde{L}(\phi^i)$ for the policy parameter $\phi^i$ of the $i$-th learner as the following *augmented* version:

$$\widetilde{L}(\phi^i) = L(\phi^i) + \mathbf{1}_{\{i \neq b\}} \beta \mathbb{E}_{s \sim \mathcal{D}} \left[ D(\pi_{\phi^i}, \pi_{\phi^b}) \right] \tag{1}$$

where $L(\phi^i)$ is the policy loss function of the base algorithm, $\mathbf{1}_{\{\cdot\}}$ denotes the indicator function, $\beta(> 0)$ is a weighting factor, $D(\pi, \pi')$ be some distance measure between two policies $\pi$ and $\pi'$.

### 3.1 THEORETICAL GUARANTEE OF MONOTONE IMPROVEMENT OF EXPECTED CUMULATIVE RETURN

In this section, we analyze the performance of the proposed soft-fusion approach theoretically and show the effectiveness of the proposed soft-fusion approach. Consider the current update period and its previous update period. Let $\pi_{\phi^i}^{old}$ be the policy of the $i$-th learner at the end of the previous update period and let $\pi_{\phi^b}$ be the best policy among all policies $\pi_{\phi^i}^{old}, i = 1, \cdots, N$. Now, consider any learner $i$ who is not the best in the previous update period. Let the policy of learner $i$ in the current update period be denoted by $\pi_{\phi^i}$, and let the policy loss function of the base algorithm be denoted as $L(\pi_{\phi^i})$. In order to analyze the performance, we consider $L(\pi_{\phi^i})$ in the form of $L(\pi_{\phi^i}) = \mathbb{E}_{s \sim \mathcal{D}, a \sim \pi_{\phi^i}(\cdot|s)} \left[ -Q^{\pi_{\phi^i}^{old}}(s, a) \right]$. The reason behind this choice is that most of actor-critic methods update the value (or Q-)function and the policy iteratively. That is, for given $\pi_{\phi^i}^{old}$, the Q-function is first updated to approximate $Q^{\pi_{\phi^i}^{old}}$. Then, with the approximation $Q^{\pi_{\phi^i}^{old}}$, the policy is

updated to yield an updated policy $\pi_{\phi^i}^{new}$. This procedure is repeated iteratively. Such loss function is used in many RL algorithms such as SAC and TD3 (Haarnoja et al. (2018); Fujimoto et al. (2018)). For the distance measure $D(\pi, \pi')$ between two policies $\pi$ and $\pi'$, we consider the KL divergence $KL(\pi||\pi')$ for analysis. Then, by eq. (1) the augmented loss function for non-best learner $i$ at the current update period is expressed as

$$\widetilde{L}(\pi_{\phi^i}) = \mathbb{E}_{s\sim\mathcal{D},a\sim\pi_{\phi^i}(\cdot|s)}\left[-Q^{\pi_{\phi^i}^{old}}(s,a)\right] + \beta\mathbb{E}_{s\sim\mathcal{D}}[KL(\pi_{\phi^i}(\cdot|s)||\pi_{\phi^b}(\cdot|s))] \tag{2}$$

$$= \mathbb{E}_{s\sim\mathcal{D}}\left[\mathbb{E}_{a\sim\pi_{\phi^i}(\cdot|s)}\left[-Q^{\pi_{\phi^i}^{old}}(s,a) + \beta\log\frac{\pi_{\phi^i}(a|s)}{\pi_{\phi^b}(a|s)}\right]\right] \tag{3}$$

Let $\pi_{\phi^i}^{new}$ be a solution that minimizes the augmented loss function eq. (3). We assume the following conditions.

**Assumption 1.** *For all $s$,*

$$\mathbb{E}_{a\sim\pi_{\phi^b}(\cdot|s)}\left[Q^{\pi_{\phi^i}^{old}}(s,a)\right] \geq \mathbb{E}_{a\sim\pi_{\phi^i}^{old}(\cdot|s)}\left[Q^{\pi_{\phi^i}^{old}}(s,a)\right]. \tag{A1}$$

**Assumption 2.** *For some $\rho, d > 0$,*

$$KL\left(\pi_{\phi^i}^{new}(\cdot|s)||\pi_{\phi^b}(\cdot|s)\right) \geq \max\left\{\rho\max_{s'}KL\left(\pi_{\phi^i}^{new}(\cdot|s')||\pi_{\phi^i}^{old}(\cdot|s')\right), d\right\}, \ \forall s. \tag{A2}$$

Assumption 1 means that if we draw the first time step action $a$ from $\pi_{\phi^b}$ and the following actions from $\pi_{\phi^i}^{old}$, then this yields better performance on average than the case that we draw all actions including the first time step action from $\pi_{\phi^i}^{old}$. This makes sense because of the definition of $\pi_{\phi^b}$. Assumption 2 is about the distance relationship among the policies to ensure a certain level of spreading of the policies for the proposed soft-fusion approach. With the two assumptions above, we have the following theorem regarding the proposed soft-fusion parallel learning scheme:

**Theorem 1.** *Under Assumptions 1 and 2, the following inequality holds:*

$$Q^{\pi_{\phi^i}^{new}}(s,a) \overset{(a)}{\geq} Q^{\pi_{\phi^b}}(s,a) + \beta\underbrace{\mathbb{E}_{s_{t+1}:s_\infty\sim\pi_{\phi^b}}\left[\sum_{k=t+1}^\infty \gamma^{k-t}\Delta(s_k)\right]}_{\textit{Improvement gap}} \overset{(b)}{\geq} Q^{\pi_{\phi^b}}(s,a) \ \forall(s,a), \ \forall i \neq b.$$

$$\tag{4}$$

*where*

$$\Delta(s) = KL\left(\pi_{\phi^i}^{new}(\cdot|s)||\pi_{\phi^b}(\cdot|s)\right) - \max\left\{\rho\max_{s'}KL\left(\pi_{\phi^i}^{new}(\cdot|s')||\pi_{\phi^i}^{old}(\cdot|s')\right), d\right\}. \tag{5}$$

*Here, inequality (a) requires Assumption 1 only and inequality (b) requires Assumption 2.*

*Proof.* See Appendix A. □

Theorem 1 states that the new solution $\pi_{\phi^i}^{new}$ for the current update period with the augmented loss function yields better performance (in the expected reward sense) than the best policy $\pi_{\phi^b}$ of the previous update period for any non-best learner $i$ of the previous update period. Hence, the proposed parallel learning scheme yields monotone improvement of expected cumulative return.

## 3.2 IMPLEMENTATION

The proposed P3S method can be applied to any off-policy base RL algorithms whether the base RL algorithms have discrete or continuous actions. For implementation, we assume that the best policy update period consists of $M$ time steps. We determine the best learner at the end of each update period based on the average of the most recent $E_r$ episodic rewards of each learner. The key point in implementation is adaptation of $\beta$ so that the improvement gap $\beta\mathbb{E}_{s_{t+1}:s_\infty\sim\pi^b}\left[\sum_{k=t+1}^\infty\gamma^{k-t}\Delta(s_k)\right]$ in (4) becomes non-negative and is maximized for given $\rho$ and $d$. The gradient of the improvement gap with respect to $\beta$ is given by $\bar{\Delta} := \mathbb{E}_{s_{t+1}:s_\infty\sim\pi^b}\left[\sum_{k=t+1}^\infty\gamma^{k-t}\Delta(s_k)\right]$, and $\bar{\Delta}$ is the average (with forgetting) of $\Delta(s_k)$ by using samples from $\pi^b$. Hence, if $\bar{\Delta} > 0$, i.e.,

$\mathrm{KL}\left(\pi_{\phi^i}^{new}(\cdot|s)||\pi_{\phi^b}(\cdot|s)\right) > \max\left\{\rho\max_{s'}\mathrm{KL}\left(\pi_{\phi^i}^{new}(\cdot|s')||\pi_{\phi^i}^{old}(\cdot|s')\right),d\right\}$ on average, then $\beta$ should be increased to maximize the performance gain. Otherwise, $\beta$ should be decreased. Therefore, we adopt the following adaptation rule for $\beta$ which is common for all non-best learners:

$$\beta = \begin{cases} \beta \leftarrow 2\beta & \text{if } \widehat{D}_{spread} > \max\{\rho\widehat{D}_{change}, d_{min}\} \times 1.5 \\ \beta \leftarrow \beta/2 & \text{if } \widehat{D}_{spread} < \max\{\rho\widehat{D}_{change}, d_{min}\}/1.5 \end{cases}. \qquad (6)$$

Here, $\widehat{D}_{spread} = \frac{1}{N-1}\sum_{i\in I^{-b}}\mathbb{E}_{s\sim\mathcal{D}}\left[D(\pi_{\phi^i}^{new}, \pi_{\phi^b})\right]$ is the estimated distance between $\pi_{\phi^i}^{new}$ and $\pi_{\phi^b}$, and $\widehat{D}_{change} = \frac{1}{N-1}\sum_{i\in I^{-b}}\mathbb{E}_{s\sim\mathcal{D}}\left[D(\pi_{\phi^i}^{new}, \pi_{\phi^i}^{old})\right]$ is the estimated distance between $\pi_{\phi^i}^{new}$ and $\pi_{\phi^i}^{old}$ averaged over all $N-1$ non-best learners, where $d_{min}$ and $\rho$ are predetermined hyperparameters. $\widehat{D}_{spread}$ and $\max\{\rho\widehat{D}_{change}, d_{min}\}$ are our practical implementations of the left-hand side (LHS) and the right-hand side (RHS) of eq. (A2), respectively. This adaptation method is similar to that used in PPO (Schulman et al. (2017)).

The update (6) of $\beta$ is done every $M$ time steps and the updated $\beta$ is used for the next $M$ time steps. As time steps elapse, $\beta$ is settled down so that $\widehat{D}_{spread}$ is around $d_{search} = \max\{\rho\widehat{D}_{change}, d_{min}\}$ and this implements Assumption 2 with equality. Hence, the proposed P3S scheme searches a spread area with rough radius $d_{search}$ around the best policy in the policy space, as illustrated in Fig. 2. The search radius $d_{search}$ is determined proportionally to $\widehat{D}_{change}$ that represents the speed of change in each learner's policy. In the case of being stuck in local optima, the change $\widehat{D}_{change}$ can be small, making the search area narrow. Hence, we set a minimum search radius $d_{min}$ to encourage escaping out of local optima.

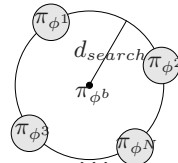

Figure 2: The conceptual search coverage in the policy space by parallel learners

We applied P3S to TD3 as the base algorithm. The constructed algorithm is named P3S-TD3. The details of TD3 is explained in Appendix G. We used the mean square difference given by $D(\pi(s), \pi'(s)) = \frac{1}{2}\|\pi(s) - \pi'(s)\|_2^2$ as the distance measure between two policies for P3S-TD3. Note that if we consider two deterministic policies as two stochastic policies with same standard deviation, the KL divergence between the two stochastic policies is the same as the mean square difference. For initial exploration P3S-TD3 uses a uniform random policy and does not update all policies over the first $T_{initial}$ time steps. The pseudocode of the P3S-TD3 is given in Appendix H. The implementation code for P3S-TD3 is available at `https://github.com/wyjung0625/p3s`.

## 4 EXPERIMENTS

### 4.1 COMPARISON TO BASELINES

In this section, we provide numerical results on performance comparison between the proposed P3S-TD3 algorithm and current state-of-the-art on-policy and off-policy baseline algorithms on several MuJoCo environments (Todorov et al. (2012)). The baseline algorithms are Proximal Policy Optimization (PPO) (Schulman et al. (2017)), Actor Critic using Kronecker-Factored Trust Region (ACKTR) (Wu et al. (2017)), Soft Q-learning (SQL) (Haarnoja et al. (2017)), (clipped double Q) Soft Actor-Critic (SAC) (Haarnoja et al. (2018)), and TD3 (Fujimoto et al. (2018)).

**Hyper-parameter setting** All hyper-parameters we used for evaluation are the same as those in the original papers (Schulman et al. (2017); Wu et al. (2017); Haarnoja et al. (2017; 2018); Fujimoto et al. (2018)). Here, we provide the hyper-parameters of the P3S-TD3 algorithm only, while details of the hyper-parameters for TD3 are provided in Appendix I. On top of the hyper-parameters for the base algorithm TD3, we used $N = 4$ learners for P3S-TD3. To update the best policy and $\beta$, the period $M = 250$ is used. The number of recent episodes $E_r = 10$ was used to determine the best learner $b$. For the search range, we used the parameter $\rho = 2$, and tuned $d_{min}$ among $d_{min} = \{0.02, 0.05\}$ for all environments. Details on $d_{min}$ for each environment is shown in Appendix I. The time steps for initial exploration $T_{initial}$ is set as 250 for Hopper-v1 and Walker2d-v1 and as 2500 for HalfCheetah-v1 and Ant-v1.

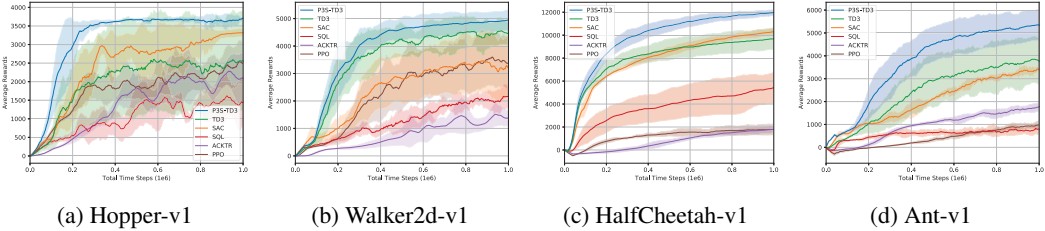

(a) Hopper-v1    (b) Walker2d-v1    (c) HalfCheetah-v1    (d) Ant-v1

Figure 3: Performance for PPO (red), ACKTR (purple), SQL (brown), (clipped double Q) SAC (orange), TD3 (green), and P3S-TD3 (proposed method, blue) on MuJoCo tasks.

**Evaluation method** Fig. 3 shows the learning curves over one million time steps for several MuJoCo tasks: Hopper-v1, Walker2d-v1, HalfCheetah-v1, and Ant-v1. In order to have sample-wise fair comparison among the considered algorithms, the time steps in the $x$-axis in Fig. 3 for P3S-TD3 is the sum of time steps of all $N$ users. For example, in the case that $N = 4$ and each learner performs 100 time steps in P3S-TD3, the corresponding $x$-axis value is 400 time steps. Since each learner performs parameter update once with one interaction with environment per each time step in P3S-TD3, the total number of parameter updates at the same $x$-axis value in Fig. 3 is the same for all algorithms including P3S-TD3, and the total number of interactions with environment at the same $x$-axis value in Fig. 3 is also the same for all algorithms including P3S-TD3. Here, the performance is obtained through the evaluation method which is similar to those in Haarnoja et al. (2018); Fujimoto et al. (2018). Evaluation of the policies is conducted every $R_{eval} = 4000$ time steps for all algorithms. At each evaluation instant, the agent (or learner) fixes its policy as the one at the evaluation instant, and interacts with the same environment separate for the evaluation purpose with the fixed policy to obtain 10 episodic rewards. The average of these 10 episodic rewards is the performance at the evaluation instant. In the case of P3S-TD3 and other parallel learning schemes, each of the $N$ learners fixes its policy as the one at the evaluation instant, and interacts with the environment with the fixed policy to obtain 10 episodic rewards. First, the 10 episodic rewards are averaged for each learner and then the maximum of the 10-episode-average rewards of the $N$ learners is taken as the performance at that evaluation instant. We performed this operation for five different random seeds, and the mean and variance of the learning curve are obtained from these five simulations. The policies used for evaluation are stochastic for PPO and ACKTR, and deterministic for the others.

**Performance on MuJoCo environments** In Fig. 3, it is observed that all baseline algorithms is similar to that in the original papers (Schulman et al. (2017); Haarnoja et al. (2018); Fujimoto et al. (2018)). With this verification, we proceed to compare P3S-TD3 with the baseline algorithms. It is seen that the P3S-TD3 algorithm outperforms the state-of-the-art RL algorithms in terms of both the speed of convergence with respect to time steps and the final steady-state performance (except in Walker2d-v1, the initial convergence is a bit slower than TD3.) Especially, in the cases of Hopper-v1 and Ant-v1, TD3 has large variance and this implies that the performance of TD3 is quite dependent on the initialization and it is not easy for TD3 to escape out of bad local minima resulting from bad initialization in certain environments. However, it is seen that P3S-TD3 yields much smaller variance than TD3. This implies that the wide area search by P3S in the policy space helps the learners escape out of bad local optima.

## 4.2 COMPARISON WITH OTHER PARALLEL LEARNING SCHEMES AND ABLATION STUDY

In the previous subsection, we observed that P3S enhances the performance and reduces dependence on initialization as compared to the single learner case with the same complexity. In fact, this should be accomplished by any properly-designed parallel learning scheme. Now, in order to demonstrate the true advantage of P3S, we compare P3S with other parallel learning schemes. P3S has several components to improve the performance based on parallelism: 1) sharing experiences from multiple policies, 2) using the best policy information, and 3) soft fusion of the best policy information for wide search area. We investigated the impact of each component on the performance improvement. For comparison we considered the following parallel policy search methods gradually incorporating more techniques:

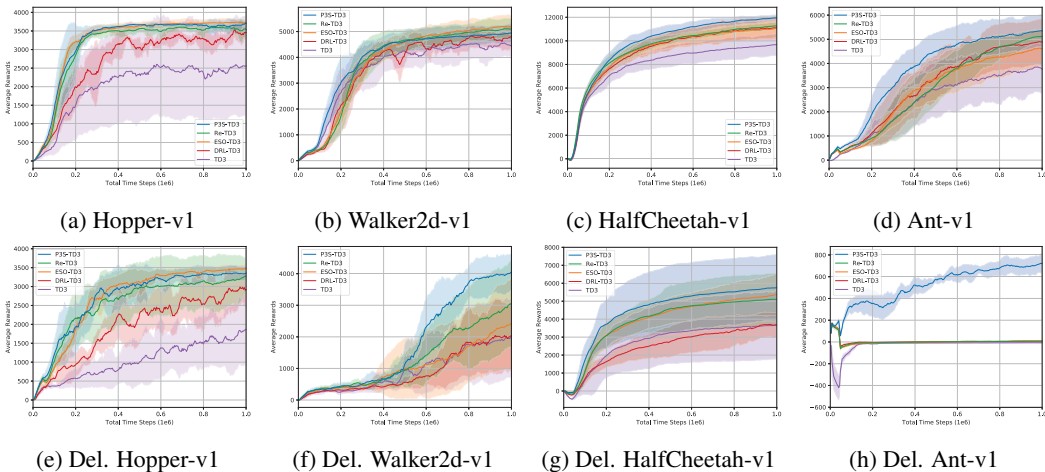

(a) Hopper-v1  (b) Walker2d-v1  (c) HalfCheetah-v1  (d) Ant-v1

(e) Del. Hopper-v1  (f) Del. Walker2d-v1  (g) Del. HalfCheetah-v1  (h) Del. Ant-v1

Figure 4: Performance of different parallel learning methods on MuJoCo environments (up), on delayed MuJoCo environments (down)

1. **Original Algorithm** The original algorithm (TD3) with one learner

2. **Distributed RL (DRL)** $N$ actors obtain samples from $N$ environment copies. The common policy and the experience replay buffer are shared by all $N$ actors.

3. **Experience-Sharing-Only (ESO)** $N$ learners interact with $N$ environment copies and update their own policies using experiences drawn from the shared experience replay buffer.

4. **Resetting (Re)** At every $M'$ time steps, the best policy is determined and all policies are initialized as the best policy, i.e., the best learner's policy parameter is copied to all other learners. The rest of the procedure is the same as experience-sharing-only algorithm.

5. **P3S** At every $M$ time steps, the best policy information is determined and this policy is used in a soft manner based on the augmented loss function.

Note that the resetting method also exploits the best policy information from $N$ learners. The main difference between P3S and the resetting method is the way the best learner's policy parameter is used. The resetting method initializes all policies with the best policy parameter every $M'$ time steps like in PBT (Jaderberg et al. (2017)), whereas P3S algorithm uses the best learner's policy parameter information determined every $M$ time steps to construct an augmented loss function. For fair comparison, $M$ and $M'$ are determined independently and optimally for P3S and Resetting, respectively, since the optimal period can be different for the two methods. We tuned $M'$ among $\{2000, 5000, 10000\}$ (MuJoCo environments) and $\{10000, 20000, 50000\}$ (Delayed MuJoCo environments) for Re-TD3, whereas $M = 250$ was used for P3S-TD3. The specific parameters used for Re-TD3 are in Appendix I. Since all $N$ policies collapse to one point in the resetting method at the beginning of each period, we expect that a larger period is required for resetting to have sufficiently spread policies at the end of each best policy selection period. We compared the performance of the aforementioned parallel learning methods combined with TD3 on two classes of tasks; MuJoCo environments, and Delayed sparse reward MuJoCo environments.

**Performance on MuJoCo environments** The upper part of Fig. 4 shows the learning curves of the considered parallel learning methods combined with TD3 for the four tasks (Hopper-v1, Walkerd-v1, HalfCheetah-v1 and Ant v1). It is seen that P3S-TD3 outperforms other parallel methods: DRL-TD3, ESO-TD3 and Re-TD3 except the case that ESO-TD3 or Re-TD3 slightly outperforms P3S-TD3 in Hopper-v1 and Walker2d-v1. In the case of Hopper-v1 and Walker2d-v1, ESO-TD3 has better final (steady-state) performance than all other parallel methods. Note that ESO-TD3 obtains most diverse experiences since the $N$ learners shares the experience replay buffer but there is no interaction among the $N$ learners until the end of training. So, it seems that this diverse experience is beneficial to Hopper-v1 and Walker2d-v1.

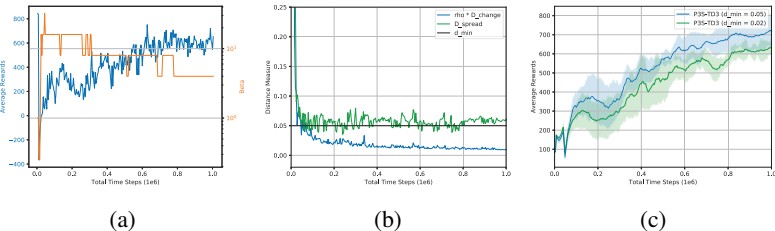

Figure 5: Ablation study of P3S-TD3 on Delayed Ant-v1: (a) Performance and $\beta$ (1 seed) with $d_{min} = 0.05$, (b) Distance measures with $d_{min} = 0.05$, and (c) Comparison with different $d_{min} = 0.02, 0.05$

**Performance on Delayed MuJoCo environments** Sparse reward environments especially require more search to obtain a good policy. To see the performance of P3S in sparse reward environments, we performed experiments on Delayed MuJoCo environments. Delayed MuJoCo environments are reward-sparsified versions of MuJoCo environments and used in Zheng et al. (2018). Delayed Mu-JoCo environments give non-zero rewards periodically with frequency $f_{reward}$ or only at the end of episodes. That is, in a delayed MuJoCo environment, the environment accumulates rewards given by the corresponding MuJoCo environment while providing zero reward to the agent, and gives the accumulated reward to the agent. We evaluated the performance on the four delayed environments with $f_{reward} = 20$: Delayed Hopper-v1, Delayed Walker2d-v1, Delayed HalfCheetah-v1 and Delayed Ant-v1.

The lower part of Fig. 4 shows the learning curves of the different parallel learning methods for the four delayed MuJoCo environments. It is seen that P3S outperforms all other considered parallel learning schemes on all environments except on delayed Hopper-v1. It seems that the enforced wide-area policy search with the soft-fusion approach in P3S is beneficial to improve performance in sparse reward environments.

**Benifits of P3S** Delayed Ant-v1 is a case of sparse reward environment in which P3S shows significant improvement as compared to other parallel schemes. As shown in Fig. 4h, the performance of TD3 drops below zero initially and converges to zero as time goes. Similar behavior is shown for other parallel methods except P3S. This is because in Delayed Ant-v1 with zero padding rewards between actual rewards, initial random actions do not generate significant positive speed to a forward direction, so it does not receive positive rewards but receives negative actual rewards due to the control cost. Once its performance less than 0, learners start learning doing nothing to reach zero reward (no positive reward and no negative reward due to no control cost). Learning beyond this seems difficult without any direction information for parameter update. This is the interpretation of the behavior of other algorithms in Fig. 4h. However, it seems that P3S escapes from this local optimum by following the best policy. This is evident in Fig. 5a, showing that after few time steps, $\beta$ is increased to follow the best policy more. Note that at the early stage of learning, the performance difference among the learners is large as seen in the large $\widehat{D}_{spread}$ values in Fig. 5b. As time elapses, all learners continue learning, the performance improves, and the spreadness among the learners' policies shrinks. However, the spreadness among the learners' policies is kept at a certain level for wide policy search by $d_{min}$, as seen in Fig. 5b. Fig. 5c shows the performance of P3S with $d_{min} = 0.05$ and $0.02$. It shows that a wide area policy search is beneficial as compared to a narrow area policy search. However, it may be detrimental to set too large a value for $d_{min}$ due to too large statistics discrepancy among samples from different learners' policies.

## 5 CONCLUSION

In this paper, we have proposed a new population-guided parallel learning scheme, P3S, to enhance the performance of off-policy RL. In the proposed P3S scheme, multiple identical learners with their own value-functions and policies sharing a common experience replay buffer search a good policy with the guidance of the best policy information in the previous search interval. The information of the best policy parameter of the previous search interval is fused in a soft manner by constructing an augmented loss function for policy update to enlarge the overall search region by the multiple

learners. The guidance by the previous best policy and the enlarged search region by P3S enables faster and better search in the policy space, and monotone improvement of expected cumulative return by P3S is theoretically proved. The P3S-TD3 algorithm constructed by applying the proposed P3S scheme to TD3 outperforms most of the current state-of-the-art RL algorithms. Furthermore, the performance gain by P3S over other parallel learning schemes is significant on harder environments especially on sparse reward environments by searching wide range in policy space.

ACKNOWLEDGMENTS

This work was supported in part by the ICT R&D program of MSIP/IITP (2016-0-00563, Research on Adaptive Machine Learning Technology Development for Intelligent Autonomous Digital Companion) and in part by the National Research Foundation of Korea(NRF) grant funded by the Korea government(Ministry of Science and ICT) (NRF2017R1E1A1A03070788).

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

## APPENDIX A. PROOF OF THEOREM 1

In this section, we prove Theorem 1. Let $\pi_{\phi^i}^{old}$ be the policy of the $i$-th learner at the end of the previous update period and let $\pi_{\phi^b}$ be the best policy among all policies $\pi_{\phi^i}^{old}, i = 1, \cdots, N$. Now, consider any learner $i$ who is not the best in the previous update period. Let the policy of learner $i$ in the current update period be denoted by $\pi_{\phi^i}$, and let the policy loss function of the base algorithm be denoted as $L(\pi_{\phi^i})$, given in the form of

$$L(\pi_{\phi^i}) = \mathbb{E}_{s \sim \mathcal{D}, a \sim \pi_{\phi^i}(\cdot|s)} \left[ -Q^{\pi_{\phi_i}^{old}}(s, a) \right]. \tag{7}$$

The reason behind this choice is that most of actor-critic methods update the value (or Q-)function and the policy iteratively. That is, for given $\pi_{\phi^i}^{old}$, the Q-function is first updated so as to approximate $Q^{\pi_{\phi^i}^{old}}$. Then, with the approximation $Q^{\pi_{\phi^i}^{old}}$ the policy is updated to yield an updated policy $\pi_{\phi^i}^{new}$, and this procedure is repeated iteratively. Such loss function is used in many RL algorithms such as SAC and TD3 (Haarnoja et al. (2018); Fujimoto et al. (2018)). SAC updates its policy by minimizing $\mathbb{E}_{s \sim \mathcal{D}, a \sim \pi'(\cdot|s)} \left[ -Q^{\pi_{old}}(s, a) + \log \pi'(a|s) \right]$ over $\pi'$. TD3 updates its policy by minimizing $\mathbb{E}_{s \sim \mathcal{D}, a = \pi'(s)} \left[ -Q^{\pi_{old}}(s, a) \right]$.

With the loss function eq. (7) and the KL divergence $KL(\pi||\pi')$ as the distance measure $D(\pi, \pi')$ between two policies $\pi$ and $\pi'$ as stated in the main paper, the augmented loss function for non-best learner $i$ at the current update period is expressed as

$$\widetilde{L}(\pi_{\phi^i}) = \mathbb{E}_{s \sim \mathcal{D}, a \sim \pi_{\phi^i}(\cdot|s)} \left[ -Q^{\pi_{\phi^i}^{old}}(s, a) \right] + \beta \mathbb{E}_{s \sim \mathcal{D}}[KL(\pi_{\phi^i}(\cdot|s)||\pi_{\phi^b}(\cdot|s))] \tag{8}$$

$$= \mathbb{E}_{s \sim \mathcal{D}} \left[ \mathbb{E}_{a \sim \pi_{\phi^i}(\cdot|s)} \left[ -Q^{\pi_{\phi^i}^{old}}(s, a) + \beta \log \frac{\pi_{\phi^i}(a|s)}{\pi_{\phi^b}(a|s)} \right] \right] \tag{9}$$

Let $\pi_{\phi^i}^{new}$ be a solution that minimizes the augmented loss function eq. (9).

**Assumption 1.** *For all $s$,*

$$\mathbb{E}_{a \sim \pi_{\phi^b}(\cdot|s)} \left[ Q^{\pi_{\phi^i}^{old}}(s, a) \right] \geq \mathbb{E}_{a \sim \pi_{\phi^i}^{old}(\cdot|s)} \left[ Q^{\pi_{\phi^i}^{old}}(s, a) \right]. \tag{10}$$

**Assumption 2.** *For some $\rho, d > 0$,*

$$KL \left( \pi_{\phi^i}^{new}(\cdot|s)||\pi_{\phi^b}(\cdot|s) \right) \geq \max \left\{ \rho \max_{s'} KL \left( \pi_{\phi^i}^{new}(\cdot|s')||\pi_{\phi^i}^{old}(\cdot|s') \right), d \right\}, \quad \forall s. \tag{11}$$

For simplicity of notations, we use the following notations from here on.

- $\pi^i$ for $\pi_{\phi^i}$
- $\pi_{old}^i$ for $\pi_{\phi^i}^{old}$
- $\pi_{new}^i$ for $\pi_{\phi^i}^{new}$
- $\pi^b$ for $\pi_{\phi^b}$.
- $KL_{max} \left( \pi_{new}^i || \pi_{old}^i \right)$ for $\max_{s'} KL \left( \pi_{\phi^i}^{new}(\cdot|s')||\pi_{\phi^i}^{old}(\cdot|s') \right)$

### A.1. A PRELIMINARY STEP

**Lemma 1.** *Let $\pi_{new}^i$ be the solution of the augmented loss function eq. (9). Then, with Assumption 1, we have the following:*

$$\mathbb{E}_{a \sim \pi_{new}^i(\cdot|s)} \left[ Q^{\pi_{old}^i}(s, a) \right] \geq \mathbb{E}_{a \sim \pi_{old}^i(\cdot|s)} \left[ Q^{\pi_{old}^i}(s, a) \right] \tag{12}$$

*for all $s$.*

*Proof.* For all $s$,

$$\mathbb{E}_{a\sim\pi_{old}^i(\cdot|s)}\left[-Q^{\pi_{old}^i}(s,a)\right] \underset{(a)}{\geq} \mathbb{E}_{a\sim\pi^b(\cdot|s)}\left[-Q^{\pi_{old}^i}(s,a)\right] \tag{13}$$

$$= \mathbb{E}_{a\sim\pi^b(\cdot|s)}\left[-Q^{\pi_{old}^i}(s,a)+\beta\log\frac{\pi^b(a|s)}{\pi^b(a|s)}\right] \tag{14}$$

$$\underset{(b)}{\geq} \mathbb{E}_{a\sim\pi_{new}^i(\cdot|s)}\left[-Q^{\pi_{old}^i}(s,a)+\beta\log\frac{\pi_{new}^i(a|s)}{\pi^b(a|s)}\right] \tag{15}$$

$$\underset{(c)}{\geq} \mathbb{E}_{a\sim\pi_{new}^i(\cdot|s)}\left[-Q^{\pi_{old}^i}(s,a)\right], \tag{16}$$

where Step (a) holds by Assumption 1, (b) holds by the definition of $\pi_{new}^i$, and (c) holds since KL divergence is always non-negative. $\square$

With Lemma 1, we prove the following preliminary result before Theorem 1:

**Proposition 1.** *With Assumption 1, the following inequality holds for all $s$ and $a$:*

$$Q^{\pi_{new}^i}(s,a) \geq Q^{\pi_{old}^i}(s,a). \tag{17}$$

*Proof of Proposition 1.* For arbitrary $s_t$ and $a_t$,

$$Q^{\pi_{old}^i}(s_t,a_t)$$

$$= r(s_t,a_t) + \gamma\mathbb{E}_{s_{t+1}\sim p(\cdot|s_t,a_t)}\left[\mathbb{E}_{a_{t+1}\sim\pi_{old}^i}\left[Q^{\pi_{old}^i}(s_{t+1},a_{t+1})\right]\right] \tag{18}$$

$$\underset{(a)}{\leq} r(s_t,a_t) + \gamma\mathbb{E}_{s_{t+1}\sim p(\cdot|s_t,a_t)}\left[\mathbb{E}_{a_{t+1}\sim\pi_{new}^i}\left[Q^{\pi_{old}^i}(s_{t+1},a_{t+1})\right]\right] \tag{19}$$

$$= \mathbb{E}_{s_{t+1}:s_{t+2}\sim\pi_{new}^i}\left[r(s_t,a_t)+\gamma r(s_{t+1},a_{t+1})+\gamma^2\mathbb{E}_{a_{t+2}\sim\pi_{old}^i}\left[Q^{\pi_{old}^i}(s_{t+2},a_{t+2})\right]\right] \tag{20}$$

$$\underset{(b)}{\leq} \mathbb{E}_{s_{t+1}:s_{t+2}\sim\pi_{new}^i}\left[r(s_t,a_t)+\gamma r(s_{t+1},a_{t+1})+\gamma^2\mathbb{E}_{a_{t+2}\sim\pi_{new}^i}\left[Q^{\pi_{old}^i}(s_{t+2},a_{t+2})\right]\right] \tag{21}$$

$$\leq \dots \tag{22}$$

$$\leq \mathbb{E}_{s_{t+1}:s_\infty\sim\pi_{new}^i}\left[\sum_{k=t}^\infty\gamma^{k-t}r(s_k,a_k)\right] \tag{23}$$

$$= Q^{\pi_{new}^i}(s_t,a_t), \tag{24}$$

where $p(\cdot|s_t,a_t)$ in eq. (18) is the environment transition probability, and $s_{t+1}:s_{t+2}\sim\pi_{new}^i$ in eq. (20) means that the trajectory from $s_{t+1}$ to $s_{t+2}$ is generated by $\pi_{new}^i$ together with the environment transition probability $p(\cdot|s_t,a_t)$. (Since the use of $p(\cdot|s_t,a_t)$ is obvious, we omitted $p(\cdot|s_t,a_t)$ for notational simplicity.) Steps (a) and (b) hold due to Lemma 1. $\square$

## A.2. PROOF OF THEOREM 1

Proposition 1 states that for a non-best learner $i$, the updated policy $\pi_{new}^i$ with the augmented loss function yields better performance than its previous policy $\pi_{old}^i$, but Theorem 1 states that for a non-best learner $i$, the updated policy $\pi_{new}^i$ with the augmented loss function yields better performance than even the previous best policy $\pi^b$.

To prove Theorem 1, we need further lemmas: We take Definition 1 and Lemma 2 directly from reference (Schulman et al. (2015)).

**Definition 1** (From Schulman et al. (2015)). *Consider two policies $\pi$ and $\pi'$. The two policies $\pi$ and $\pi'$ are $\alpha$-coupled if $Pr(a\neq a')\leq\alpha$, $(a,a')\sim(\pi(\cdot|s),\pi'(\cdot|s))$ for all $s$.*

**Lemma 2** (From Schulman et al. (2015)). *Given $\alpha$-coupled policies $\pi$ and $\pi'$, for all $s$,*

$$|\mathbb{E}_{a\sim\pi'}[A^\pi(s,a)]| \leq 2\alpha\max_{s,a}|A^\pi(s,a)|, \tag{25}$$

*where $A^\pi(s,a)$ is the advantage function.*

*Proof.* (From Schulman et al. (2015))

$$\left|\mathbb{E}_{a\sim\pi'}\left[A^\pi(s,a)\right]\right| \underset{(a)}{=} \left|\mathbb{E}_{a'\sim\pi'}\left[A^\pi(s,a')\right] - \mathbb{E}_{a\sim\pi}\left[A^\pi(s,a)\right]\right| \tag{26}$$

$$= \left|\mathbb{E}_{(a,a')\sim(\pi,\pi')}\left[A^\pi(s,a') - A^\pi(s,a)\right]\right| \tag{27}$$

$$= |\Pr(a=a')\mathbb{E}_{(a,a')\sim(\pi,\pi')|a=a'}\left[A^\pi(s,a') - A^\pi(s,a)\right]$$
$$+ \Pr(a\neq a')\mathbb{E}_{(a,a')\sim(\pi,\pi')|a\neq a'}\left[A^\pi(s,a') - A^\pi(s,a)\right]| \tag{28}$$

$$= \Pr(a\neq a')|\mathbb{E}_{(a,a')\sim(\pi,\pi')|a\neq a'}\left[A^\pi(s,a') - A^\pi(s,a)\right]| \tag{29}$$

$$\leq 2\alpha \max_{s,a}|A^\pi(s,a)|, \tag{30}$$

where Step (a) holds since $\mathbb{E}_{a\sim\pi}[A^\pi(s,a)] = 0$ for all $s$ by the property of an advantage function. $\square$

By modifying the result on the state value function in Schulman et al. (2015), we have the following lemma on the Q-function:

**Lemma 3.** *Given two policies $\pi$ and $\pi'$, the following equality holds for arbitrary $s_0$ and $a_0$:*

$$Q^{\pi'}(s_0,a_0) = Q^\pi(s_0,a_0) + \gamma\mathbb{E}_{\tau\sim\pi'}\left[\sum_{t=1}^\infty \gamma^{t-1}A^\pi(s_t,a_t)\right], \tag{31}$$

*where $\mathbb{E}_{\tau\sim\pi'}$ is expectation over trajectory $\tau$ which start from a state $s_1$ drawn from the transition probability $p(\cdot|s_0,a_0)$ of the environment.*

*Proof.* Note that

$$Q^\pi(s_0,a_0) = r_0 + \gamma\mathbb{E}_{s_1\sim p(\cdot|s_0,a_0)}\left[V^\pi(s_1)\right] \tag{32}$$

$$Q^{\pi'}(s_0,a_0) = r_0 + \gamma\mathbb{E}_{s_1\sim p(\cdot|s_0,a_0)}\left[V^{\pi'}(s_1)\right] \tag{33}$$

Hence, it is sufficient to show the following equality:

$$\mathbb{E}_{\tau\sim\pi'}\left[\sum_{t=1}^\infty \gamma^{t-1}A^\pi(s_t,a_t)\right] = \mathbb{E}_{s_1\sim p(\cdot|s_0,a_0)}\left[V^{\pi'}(s_1)\right] - \mathbb{E}_{s_1\sim p(\cdot|s_0,a_0)}\left[V^\pi(s_1)\right] \tag{34}$$

Note that

$$A^\pi(s_t,a_t) = \mathbb{E}_{s_{t+1}\sim p(\cdot|s_t,a_t)}\left[r_t + \gamma V^\pi(s_{t+1}) - V^\pi(s_t)\right] \tag{35}$$

Then, substituting eq. (35) into the LHS of eq. (34), we have

$$\mathbb{E}_{\tau\sim\pi'}\left[\sum_{t=1}^\infty \gamma^{t-1}A^\pi(s_t,a_t)\right] = \mathbb{E}_{\tau\sim\pi'}\left[\sum_{t=1}^\infty \gamma^{t-1}\left(r_t + \gamma V^\pi(s_{t+1}) - V^\pi(s_t)\right)\right] \tag{36}$$

$$= \mathbb{E}_{\tau\sim\pi'}\left[\sum_{t=1}^\infty \gamma^{t-1}r_t\right] - \mathbb{E}_{s_1\sim p(\cdot|s_0,a_0)}\left[V^\pi(s_1)\right] \tag{37}$$

$$= \mathbb{E}_{s_1\sim p(\cdot|s_0,a_0)}\left[V^{\pi'}(s_1)\right] - \mathbb{E}_{s_1\sim p(\cdot|s_0,a_0)}\left[V^\pi(s_1)\right], \tag{38}$$

where eq. (37) is valid since $\mathbb{E}_{\tau\sim\pi'}\left[\sum_{t=1}^\infty \gamma^{t-1}\left(\gamma V^\pi(s_{t+1}) - V^\pi(s_t)\right)\right] = -\mathbb{E}_{s_1\sim p(\cdot|s_0,a_0)}\left[V^\pi(s_1)\right]$. Since the RHS of eq. (38) is the same as the RHS of eq. (34), the claim holds. $\square$

Then, we can prove the following lemma regarding the difference between the Q-functions of two $\alpha$-coupled policies $\pi$ and $\pi'$:

**Lemma 4.** *Let $\pi$ and $\pi'$ be $\alpha$-coupled policies. Then,*

$$\left|Q^\pi(s,a) - Q^{\pi'}(s,a)\right| \leq \frac{2\epsilon\gamma}{1-\gamma}\max\left\{C\alpha^2, 1/C\right\}, \tag{39}$$

*where $\epsilon = \max_{s,a}|A^\pi(s,a)|$ and $C > 0$*

*Proof.* From Lemma 3, we have

$$Q^{\pi'}(s_0, a_0) - Q^\pi(s_0, a_0) = \gamma \mathbb{E}_{\tau \sim \pi'} \left[ \sum_{t=1}^\infty \gamma^{t-1} A^\pi(s_t, a_t) \right]. \tag{40}$$

Then, from eq. (40) we have

$$\left| Q^{\pi'}(s_0, a_0) - Q^\pi(s_0, a_0) \right| = \left| \gamma \mathbb{E}_{\tau \sim \pi'} \left[ \sum_{t=1}^\infty \gamma^{t-1} A^\pi(s_t, a_t) \right] \right| \tag{41}$$

$$\leq \gamma \sum_{t=1}^\infty \gamma^{t-1} \left| \mathbb{E}_{s_t, a_t \sim \pi'} \left[ A^\pi(s_t, a_t) \right] \right| \tag{42}$$

$$\leq \gamma \sum_{t=1}^\infty \gamma^{t-1} 2\alpha \max_{s,a} |A^\pi(s, a)| \tag{43}$$

$$= \frac{\epsilon \gamma}{1 - \gamma} 2\alpha \tag{44}$$

$$\leq \frac{\epsilon \gamma}{1 - \gamma} \left( C\alpha^2 + 1/C \right) \tag{45}$$

$$\leq \frac{\epsilon \gamma}{1 - \gamma} 2 \max \left\{ C\alpha^2, 1/C \right\}, \tag{46}$$

where $\epsilon = \max_{s,a} |A^\pi(s, a)|$ and $C > 0$. Here, eq. (43) is valid due to Lemma 2, eq. (45) is valid since $C\alpha^2 + 1/C - 2\alpha = C \left( \alpha - \frac{1}{C} \right)^2 \geq 0$, and eq. (46) is valid since the sum of two terms is less than or equal to two times the maximum of the two terms. $\qquad \square$

Up to now, we considered some results valid for given two $\alpha$-coupled policies $\pi$ and $\pi'$. On the other hand, it is shown in Schulman et al. (2015) that for arbitrary policies $\pi$ and $\pi'$, if we take $\alpha$ as the maximum (over $s$) of the total variation divergence $\max_s D_{TV}(\pi(\cdot|s)||\pi'(\cdot|s))$ between $\pi(\cdot|s)$ and $\pi'(\cdot|s)$, then the two policies are $\alpha$-coupled with the $\alpha$ value of $\alpha = \max_s D_{TV}(\pi(\cdot|s)||\pi'(\cdot|s))$.

Applying the above facts, we have the following result regarding $\pi_{new}^i$ and $\pi_{old}^i$:

**Lemma 5.** *For some constants $\rho, d > 0$,*

$$Q^{\pi_{new}^i}(s, a) \leq Q^{\pi_{old}^i}(s, a) + \beta \max \left\{ \rho KL_{max} \left( \pi_{new}^i || \pi_{old}^i \right), d \right\} \tag{47}$$

*for all $s$ and $a$, where $KL_{max}(\pi||\pi') = \max_s KL(\pi(\cdot|s)||\pi'(\cdot|s))$.*

*Proof.* For $\pi_{new}^i$ and $\pi_{old}^i$, take $\alpha$ as the maximum of the total variation divergence between $\pi_{new}^i$ and $\pi_{old}^i$, i.e., $\alpha = \max_s D_{TV}(\pi_{new}^i(\cdot|s)||\pi_{old}^i(\cdot|s))$. Let this $\alpha$ value be denoted by $\hat{\alpha}$. Then, by the result of Schulman et al. (2015) mentioned in the above, $\pi_{new}^i$ and $\pi_{old}^i$ are $\hat{\alpha}$-coupled with $\hat{\alpha} = \max_s D_{TV}(\pi_{new}^i(\cdot|s)||\pi_{old}^i(\cdot|s))$. Since

$$D_{TV}(\pi_{new}^i(\cdot|s)||\pi_{old}^i(\cdot|s))^2 \leq KL(\pi_{new}^i(\cdot|s)||\pi_{old}^i(\cdot|s)), \tag{48}$$

by the relationship between the total variation divergence and the KL divergence, we have

$$\hat{\alpha}^2 \leq \max_s KL(\pi_{new}^i(\cdot|s)||\pi_{old}^i(\cdot|s)). \tag{49}$$

Now, substituting $\pi = \pi_{new}^i$, $\pi' = \pi_{old}^i$ and $\alpha = \hat{\alpha}$ into eq. (39) and applying eq. (49), we have

$$\left| Q^{\pi_{new}^i}(s, a) - Q^{\pi_{old}^i}(s, a) \right| \leq \beta \max \left\{ \rho KL_{max} \left( \pi_{new}^i || \pi_{old}^i \right), d \right\} \tag{50}$$

for some $\rho, d > 0$. Here, proper scaling due to the introduction of $\beta$ is absorbed into $\rho$ and $d$. That is, $\rho$ can be set as $\frac{2\epsilon\gamma C}{\beta(1-\gamma)}$ and $d$ can be set as $\frac{2\epsilon\gamma}{\beta(1-\gamma)C}$. Then, by Proposition 1, $\left| Q^{\pi_{new}^i}(s, a) - Q^{\pi_{old}^i}(s, a) \right|$ in the LHS of eq. (50) becomes $\left| Q^{\pi_{new}^i}(s, a) - Q^{\pi_{old}^i}(s, a) \right| = Q^{\pi_{new}^i}(s, a) - Q^{\pi_{old}^i}(s, a)$. Hence, from this fact and eq. (50), we have eq. (47). This concludes proof. $\qquad \square$

**Proposition 2.** *With Assumption 1, we have*

$$\mathbb{E}_{a \sim \pi_{new}^i(\cdot|s)}\left[Q^{\pi_{new}^i}(s,a)\right] \geq \mathbb{E}_{a \sim \pi^b(\cdot|s)}\left[Q^{\pi_{new}^i}(s,a)\right] + \beta\Delta(s). \tag{51}$$

*where*

$$\Delta(s) = \left[KL\left(\pi_{new}^i(\cdot|s)||\pi^b(\cdot|s)\right) - \max\left\{\rho KL_{max}\left(\pi_{new}^i||\pi_{old}^i\right), d\right\}\right] \tag{52}$$

*Proof.*

$$\mathbb{E}_{a \sim \pi^b(\cdot|s)}\left[-Q^{\pi_{new}^i}(s,a)\right] \tag{53}$$

$$\underset{(a)}{\geq} \mathbb{E}_{a \sim \pi^b(\cdot|s)}\left[-Q^{\pi_{old}^i}(s,a)\right] - \beta\max\left\{\rho KL_{max}\left(\pi_{new}^i||\pi_{old}^i\right), d\right\} \tag{54}$$

$$= \mathbb{E}_{a \sim \pi^b(\cdot|s)}\left[-Q^{\pi_{old}^i}(s,a) + \beta\log\frac{\pi^b(a|s)}{\pi^b(a|s)}\right] - \beta\max\left\{\rho KL_{max}\left(\pi_{new}^i||\pi_{old}^i\right), d\right\} \tag{55}$$

$$\underset{(b)}{\geq} \mathbb{E}_{a \sim \pi_{new}^i(\cdot|s)}\left[-Q^{\pi_{old}}(s,a) + \beta\log\frac{\pi_{new}^i(a|s)}{\pi^b(a|s)}\right] - \beta\max\left\{\rho KL_{max}\left(\pi_{new}^i||\pi_{old}^i\right), d\right\} \tag{56}$$

$$= \mathbb{E}_{a \sim \pi_{new}^i(\cdot|s)}\left[-Q^{\pi_{old}}(s,a)\right] + \beta KL\left(\pi_{new}^i(\cdot|s)||\pi^b(\cdot|s)\right) - \beta\max\left\{\rho KL_{max}\left(\pi_{new}^i||\pi_{old}^i\right), d\right\} \tag{57}$$

$$= \mathbb{E}_{a \sim \pi_{new}^i(\cdot|s)}\left[-Q^{\pi_{old}^i}(s,a)\right] + \beta\left[KL\left(\pi_{new}^i(\cdot|s)||\pi^b(\cdot|s)\right) - \max\left\{\rho KL_{max}\left(\pi_{new}^i||\pi_{old}^i\right), d\right\}\right] \tag{58}$$

$$= \mathbb{E}_{a \sim \pi_{new}^i(\cdot|s)}\left[-Q^{\pi_{old}^i}(s,a)\right] + \beta\Delta(s) \tag{59}$$

$$\underset{(c)}{\geq} \mathbb{E}_{a \sim \pi_{new}^i(\cdot|s)}\left[-Q^{\pi_{new}^i}(s,a)\right] + \beta\Delta(s), \tag{60}$$

where step (a) is valid due to Lemma 5, step (b) is valid due to the definition of $\pi_{new}^i$, and step (c) is valid due to Proposition 1. $\qquad\square$

Finally, we prove Theorem 1.

**Theorem 1.** *Under Assumptions 1 and 2, the following inequality holds:*

$$Q^{\pi_{new}^i}(s,a) \geq Q^{\pi^b}(s,a) + \beta\mathbb{E}_{s_{t+1}:s_\infty \sim \pi^b}\left[\sum_{k=t+1}^{\infty}\gamma^{k-t}\Delta(s_k)\right] \geq Q^{\pi^b}(s,a), \quad \forall(s,a), \forall i \neq b. \tag{61}$$

*where*

$$\Delta(s) = \left[KL\left(\pi_{new}^i(\cdot|s)||\pi^b(\cdot|s)\right) - \max\left\{\rho KL_{max}\left(\pi_{new}^i||\pi_{old}^i\right), d\right\}\right] \tag{62}$$

*Proof of Theorem 1:* Proof of Theorem 1 is by recursive application of Proposition 2. For arbitrary $s_t$ and $a_t$,

$$Q^{\pi_{new}^i}(s_t, a_t)$$

$$= r(s_t, a_t) + \gamma\mathbb{E}_{s_{t+1} \sim p(\cdot|s_t, a_t)}\left[\mathbb{E}_{a_{t+1} \sim \pi_{new}^i}\left[Q^{\pi_{new}^i}(s_{t+1}, a_{t+1})\right]\right] \tag{63}$$

$$\underset{(a)}{\geq} r(s_t, a_t) + \gamma\mathbb{E}_{s_{t+1} \sim p(\cdot|s_t, a_t)}\left[\mathbb{E}_{a_{t+1} \sim \pi^b}\left[Q^{\pi_{new}^i}(s_{t+1}, a_{t+1})\right] + \beta\Delta(s_{t+1})\right] \tag{64}$$

$$= \mathbb{E}_{s_{t+1}:s_{t+2} \sim \pi^b}\left[r(s_t, a_t) + \gamma r(s_{t+1}, a_{t+1}) + \gamma^2\mathbb{E}_{a_{t+2} \sim \pi_{new}^i}\left[Q^{\pi_{new}^i}(s_{t+2}, a_{t+2})\right]\right]$$
$$+ \beta\mathbb{E}_{s_{t+1}:s_{t+2} \sim \pi^b}\left[\gamma\Delta(s_{t+1})\right] \tag{65}$$

$$\underset{(b)}{\geq} \mathbb{E}_{s_{t+1}:s_{t+2} \sim \pi^b}\left[r(s_t, a_t) + \gamma r(s_{t+1}, a_{t+1}) + \gamma^2\mathbb{E}_{a_{t+2} \sim \pi^b}\left[Q^{\pi_{new}^i}(s_{t+2}, a_{t+2})\right] + \beta\gamma^2\Delta(s_{t+2})\right]$$
$$+ \beta\mathbb{E}_{s_{t+1}:s_{t+2} \sim \pi^b}\left[\gamma\Delta(s_{t+1})\right] \tag{66}$$

$$\geq \ldots$$

$$\geq \mathbb{E}_{s_{t+1}:s_\infty \sim \pi^b} \left[ \sum_{k=t}^{\infty} \gamma^{k-t} r(s_k, a_k) \right] + \beta \mathbb{E}_{s_{t+1}:s_\infty \sim \pi^b} \left[ \sum_{k=t+1}^{\infty} \gamma^{k-t} \Delta(s_k) \right] \tag{67}$$

$$= Q^{\pi^b}(s_t, a_t) + \beta \mathbb{E}_{s_{t+1}:s_\infty \sim \pi^b} \left[ \sum_{k=t+1}^{\infty} \gamma^{k-t} \Delta(s_k) \right] \tag{68}$$

where steps (a) and (b) hold because of Proposition 2. Assumption 2 ensures

$$\Delta(s) = \left[ \mathrm{KL}\left( \pi_{new}^i(\cdot|s) || \pi^b(\cdot|s) \right) - \max\left\{ \rho \mathrm{KL}_{max}\left( \pi_{new}^i || \pi_{old}^i \right), d \right\} \right] \geq 0, \quad \forall s. \tag{69}$$

Hence, the second term in (68) is non-negative. Therefore, we have

$$Q^{\pi_{new}^i}(s_t, a_t) \quad \geq \quad Q^{\pi^b}(s_t, a_t) + \beta \mathbb{E}_{s_{t+1}:s_\infty \sim \pi^b} \left[ \sum_{k=t+1}^{\infty} \gamma^{k-t} \Delta(s_k) \right] \tag{70}$$

$$\geq \quad Q^{\pi^b}(s_t, a_t) \tag{71}$$

$$\square$$

APPENDIX B. INTUITION OF THE IMPLEMENTATION OF $\beta$ ADAPTATION

Due to Theorem 1, we have

$$Q^{\pi_{new}^i}(s_t, a_t) \geq Q^{\pi^b}(s_t, a_t) + \underbrace{\beta \mathbb{E}_{s_{t+1}:s_\infty \sim \pi^b}\left[\sum_{k=t+1}^{\infty} \gamma^{k-t}\Delta(s_k)\right]}_{\text{Improvement gap}} \qquad (72)$$

where

$$\Delta(s) = \left[\text{KL}\left(\pi_{new}^i(\cdot|s)||\pi^b(\cdot|s)\right) - \max\left\{\rho \text{KL}_{max}\left(\pi_{new}^i||\pi_{old}^i\right), d\right\}\right] \geq 0, \quad \forall s. \qquad (73)$$

In deriving eqs. (72) and (73), we only used Assumption 1. When we have Assumption 2, the improvement gap term in (72) becomes non-negative and we have

$$Q^{\pi_{new}^i}(s_t, a_t) \geq Q^{\pi^b}(s_t, a_t) \qquad (74)$$

as desired. However, in practice, Assumption 2 should be implemented so that the improvement gap term becomes non-negative and we have the desired result (74). The implementation of the condition is through adaptation of $\beta$. We adapt $\beta$ to maximize the improvement gap $\beta \mathbb{E}_{s_{t+1}:s_\infty \sim \pi^b}\left[\sum_{k=t+1}^{\infty} \gamma^{k-t}\Delta(s_k)\right]$ in (72) for given $\rho$ and $d$. Let us denote $\mathbb{E}_{s_{t+1}:s_\infty \sim \pi^b}\left[\sum_{k=t+1}^{\infty} \gamma^{k-t}\Delta(s_k)\right]$ by $\bar{\Delta}$. Then, the improvement gap is given by $\beta\bar{\Delta}$. Note that $\bar{\Delta}$ is the average (with forgetting) of $\Delta(s_k)$ by using samples from $\pi^b$. The gradient of the improvement gap with respect to $\beta$ is given by

$$\nabla_\beta(\beta\bar{\Delta}) = \bar{\Delta}. \qquad (75)$$

Thus, if $\bar{\Delta} > 0$, i.e., $\text{KL}\left(\pi_{new}^i(\cdot|s)||\pi^b(\cdot|s)\right) > \max\left\{\rho \text{KL}_{max}\left(\pi_{new}^i||\pi_{old}^i\right), d\right\}$ on average, then $\beta$ should be increased to maximize the performance gain. On the other hand, if $\bar{\Delta} < 0$, i.e., $\text{KL}\left(\pi_{new}^i(\cdot|s)||\pi^b(\cdot|s)\right) \leq \max\left\{\rho \text{KL}_{max}\left(\pi_{new}^i||\pi_{old}^i\right), d\right\}$ on average, then $\beta$ should be decreased. Therefore, we adapt $\beta$ as follows:

$$\beta = \begin{cases} \beta \leftarrow 2\beta & \text{if } \widehat{D}_{spread} > \max\{\rho\widehat{D}_{change}, d_{min}\} \times 1.5 \\ \beta \leftarrow \beta/2 & \text{if } \widehat{D}_{spread} < \max\{\rho\widehat{D}_{change}, d_{min}\}/1.5 \end{cases}. \qquad (76)$$

where $\widehat{D}_{spread}$ and $\widehat{D}_{change}$ are implementations of $\text{KL}\left(\pi_{new}^i(\cdot|s)||\pi^b(\cdot|s)\right)$ and $\text{KL}_{max}\left(\pi_{new}^i||\pi_{old}^i\right)$, respectively.

## APPENDIX C. COMPARISON TO BASELINES ON DELAYED MUJOCO ENVIRONMENTS

In this section, we provide the learning curves of the state-of-the-art single-learner baselines on delayed MuJoCo environments. Fig. 6 shows the learning curves of P3S-TD3 algorithm and the single-learner baselines on the four delayed MuJoCo environments: Delayed Hopper-v1, Delayed Walker2d-v1, Delayed HalfCheetah-v1, and Delayed Ant-v1. It is observed that in the delayed Ant-v1 environment, ACKTR outperforms the P3S-TD3 algorithm. It is also observed that P3S-TD3 significantly outperforms all baselines on all other environments than the delayed Ant-v1 environment.

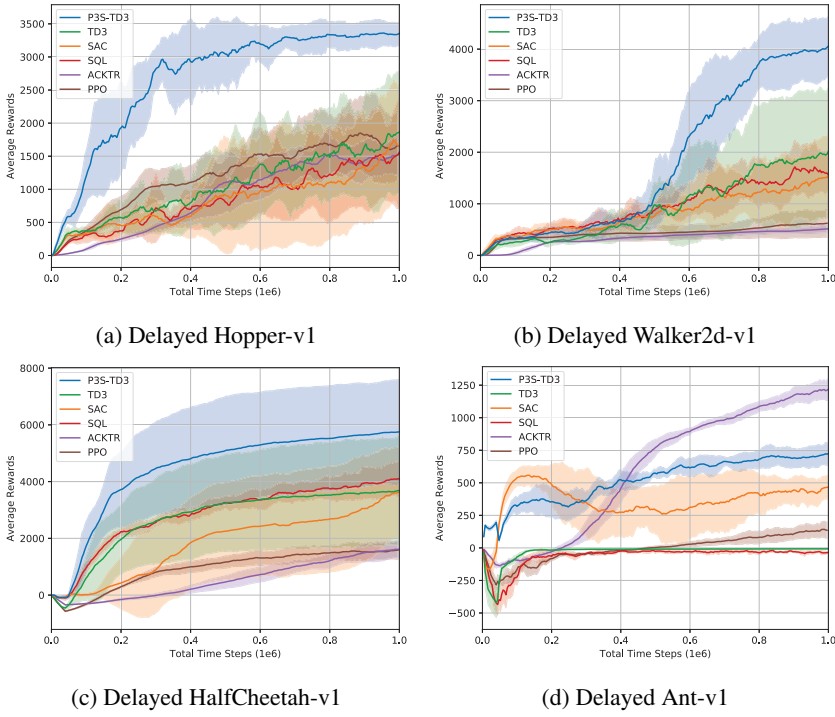

(a) Delayed Hopper-v1          (b) Delayed Walker2d-v1

(c) Delayed HalfCheetah-v1        (d) Delayed Ant-v1

Figure 6: Performance for PPO (brown), ACKTR (purple), (clipped double Q) SAC (orange), TD3 (green), and P3S-TD3 (proposed method, blue) on the four delayed MuJoCo tasks with $f_{reward} = 20$.

## Appendix D. Comparison to CEM-TD3

In this section, we compare the performance of TD3 and P3S-TD3 with CEM-TD3 (Pourchot & Sigaud (2019)), which is a state-of-the-art evolutionary algorithm. CEM-TD3 uses a population to search a better policy as other evolutionary algorithms do. The operation of CEM-TD3 is described as follows:

1. It first samples $N$ policies by drawing policy parameters from a Gaussian distribution.
2. It randomly selects a half of the population. The selected policies and a common Q function are updated based on minibatches drawn from a common replay buffer.
3. Both the updated selected policies and the unselected policies are evaluated and the experiences during the evaluation are stored in the common replay buffer.
4. After evaluation of all $N$ policies, it takes the best $N/2$ policies and updates the mean and variance of the policy parameter distribution as the mean and variance of the parameters of the best $N/2$ policies.
5. Steps 1 to 4 are iterated until the time steps reach maximum.

For the performance comparision, we used the implementation of CEM-TD3 in the original paper (Pourchot & Sigaud (2019))[1] with the hyper-parameters provided therein. Table 1 shows the steady state performance on MuJoCo and delayed MuJoCo environments. Fig. 7 in the next page shows the learning curves on MuJoCo and delayed MuJoCo environments. It is seen that P3S-TD3 outperforms CEM-TD3 on all environments except delayed HalfCheetah-v1. Notice that P3S-TD3 significantly outperforms CEM-TD3 on delayed Walker2d-v1 and delayed Ant-v1.

Table 1: Steady state performance of P3S-TD3, CEM-TD3, and TD3

| Environment | P3S-TD3 | CEM-TD3 | TD3 |
|---|---|---|---|
| Hopper-v1 | **3705.92** | 3686.08 | 2555.85 |
| Walker2d-v1 | **4953.00** | 4819.40 | 4455.51 |
| HalfCheetah-v1 | **11961.44** | 11417.73 | 9695.92 |
| Ant-v1 | **5339.66** | 4379.73 | 3760.50 |
| Delayed Hopper-v1 | **3355.53** | 3117.20 | 1866.02 |
| Delayed Walker2d-v1 | **4058.85** | 1925.63 | 2016.48 |
| Delayed HalfCheetah-v1 | 5754.80 | **6389.40** | 3684.28 |
| Delayed Ant-v1 | **724.50** | 70.44 | -7.45 |

---

[1] The code is available at https://github.com/apourchot/CEM-RL.

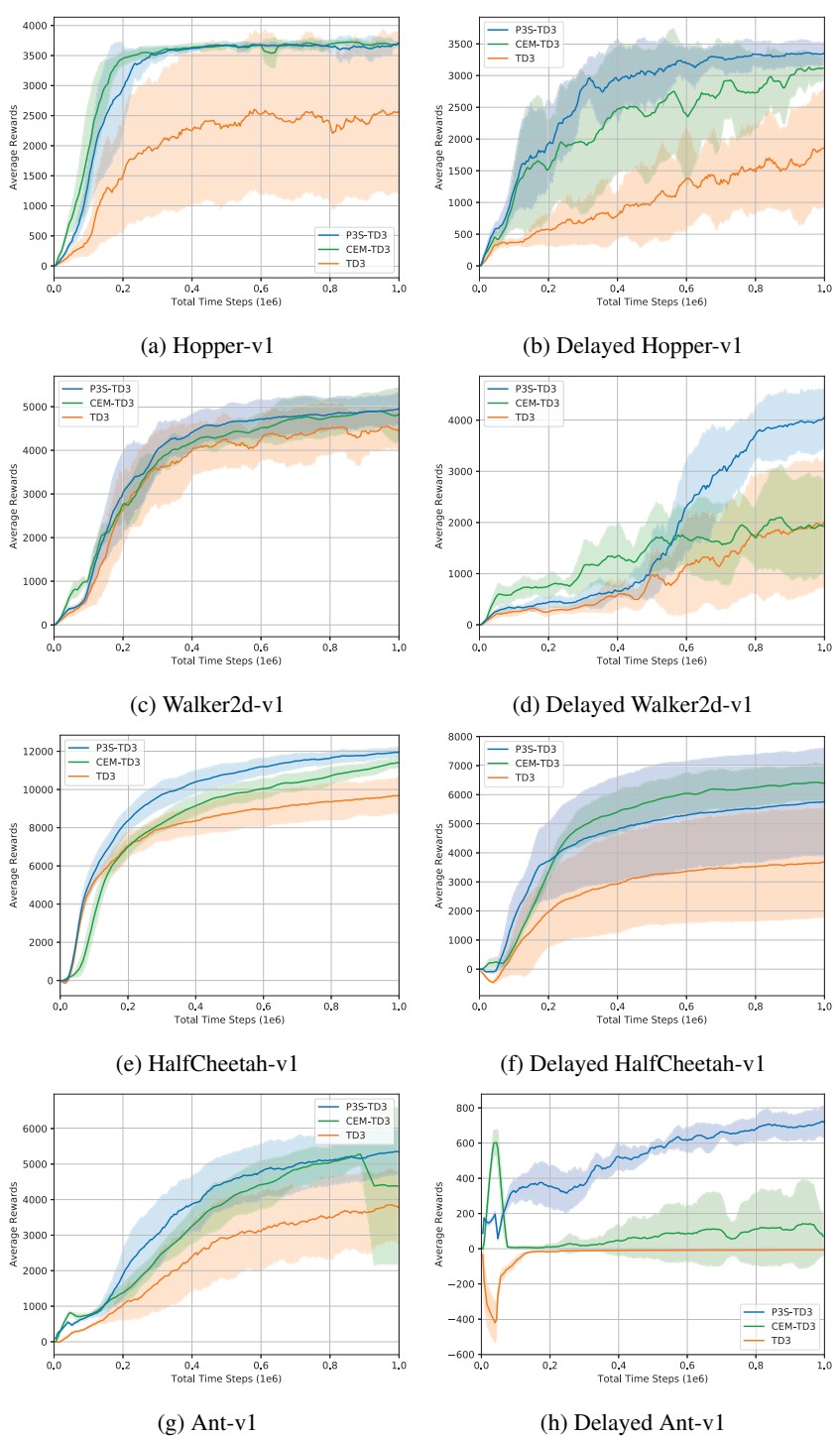

(a) Hopper-v1

(b) Delayed Hopper-v1

(c) Walker2d-v1

(d) Delayed Walker2d-v1

(e) HalfCheetah-v1

(f) Delayed HalfCheetah-v1

(g) Ant-v1

(h) Delayed Ant-v1

Figure 7: Performance of P3S-TD3, CEM-TD3, and TD3

APPENDIX E. COMPARISON TO METHOD USING CENTER POLICY

In this section, we consider a variant of the proposed P3S-TD3 algorithm, named Center-TD3. This variant uses a center policy like in Distral (Teh et al. (2017)) and Divide-and-Conquer (Ghosh et al. (2018)). Center-TD3 has $N$ policies and a center policy $\pi^c$ in addition. The value and policy parameter update procedure is the same as the original TD3 algorithm but the loss functions are newly defined. That is, the Q function loss is the same as the original TD3 algorithm, but the parameters of the $N$ policies are updated based on the following loss:

$$\tilde{L}(\phi^i) = \hat{\mathbb{E}}_{s \sim \mathcal{D}} \left[ -Q_{\theta_1^i}(s, \pi_{\phi^i}(s)) + \frac{\beta}{2} \left\| \pi_{\phi^i}(s) - \pi_{\phi^c}(s) \right\|_2^2 \right]. \tag{77}$$

The parameter loss function of Center-TD3 is obtained by replacing the best policy with the center policy in the parameter loss function of P3S-TD3. The center policy is updated in every $M$ time steps to the direction of minimizing the following loss

$$\tilde{L}(\phi^c) = \hat{\mathbb{E}}_{s \sim \mathcal{D}} \left[ \frac{\beta}{2} \sum_{i=1}^{N} \left\| \pi_{\phi^i}(s) - \pi_{\phi^c}(s) \right\|_2^2 \right]. \tag{78}$$

Center-TD3 follows the spirit of Distral (Teh et al. (2017)) and Divide-and-Conquer (Ghosh et al. (2018)) algorithms.

We tuned and selected the hyper-parameters for Center-TD3 from the sets $\beta \in \{1, 10\}$ and $M \in \{2, 20, 40, 100, 200, 500\}$. We then measured the performance of Center-TD3 by using the selected hyper-parameter $\beta = 1$, $M = 40$. Fig. 8 in the next page and Table 2 show the learning curves and the steady-state performance on MuJoCo and delayed MuJoCo environments, respectively. It is seen that P3S-TD3 outperforms Center-TD3 on all environments except Delayed HalfCheetah-v1.

Table 2: Steady state performance of P3S-TD3, Center-TD3, and TD3

| Environment | P3S-TD3 | Center-TD3 | TD3 |
|---|---|---|---|
| Hopper-v1 | **3705.92** | 3675.28 | 2555.85 |
| Walker2d-v1 | **4953.00** | 4689.34 | 4455.51 |
| HalfCheetah-v1 | **11961.44** | 10620.84 | 9695.92 |
| Ant-v1 | **5339.66** | 4616.82 | 3760.50 |
| Delayed Hopper-v1 | **3355.53** | 3271.50 | 1866.02 |
| Delayed Walker2d-v1 | **4058.85** | 2878.85 | 2016.48 |
| Delayed HalfCheetah-v1 | 5754.80 | **6047.47** | 3684.28 |
| Delayed Ant-v1 | **724.50** | 688.96 | -7.45 |

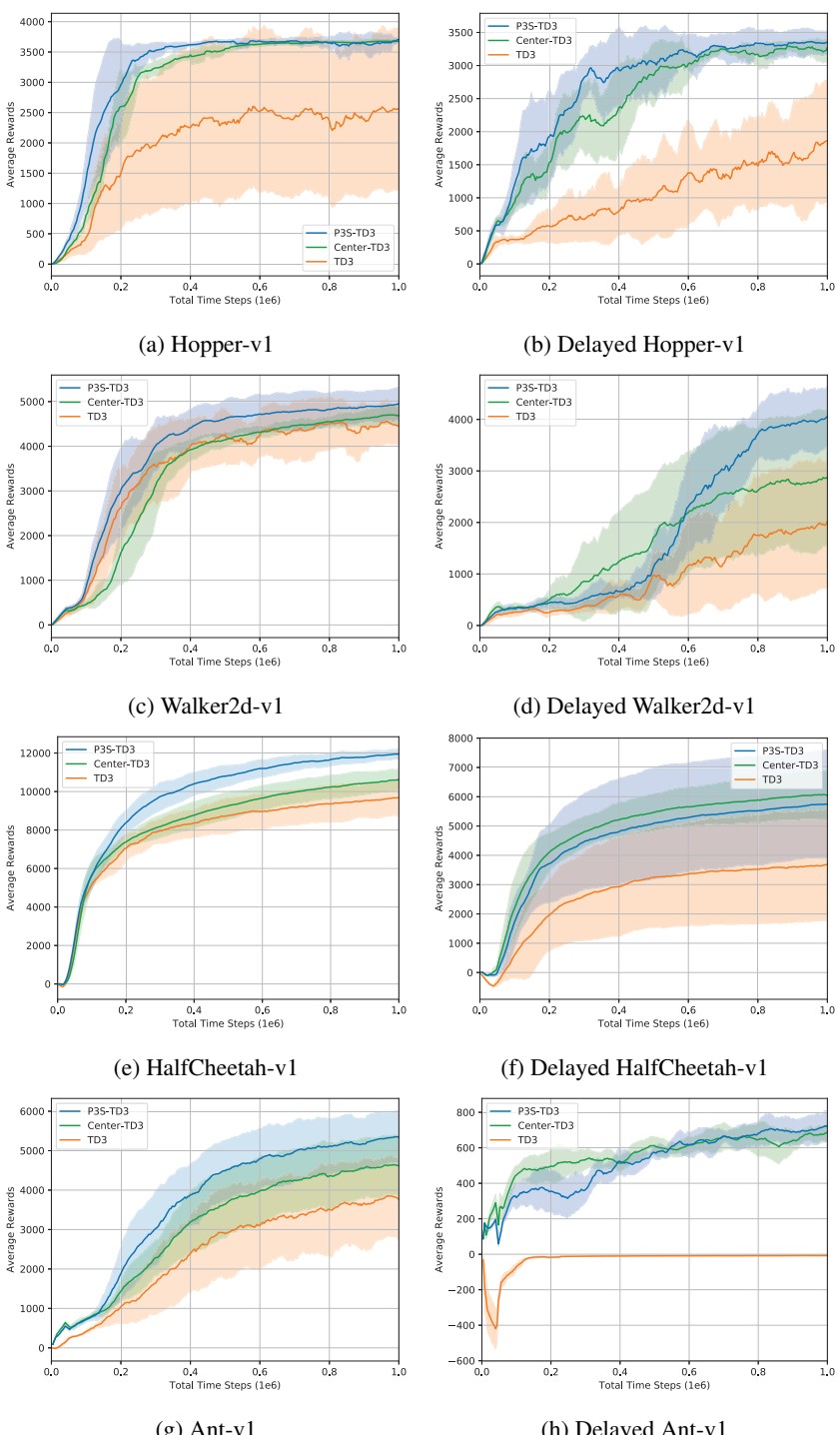

Figure 8: Performance of P3S-TD3, Center-TD3, and TD3

## APPENDIX F. RESULT ON SWIMMER-V1

Khadka & Tumer (2018); Pourchot & Sigaud (2019) noticed that most deep RL methods suffer from a deceptive gradient problem on the Swimmer-v1 task, and most RL methods could not learn effectively on the Swimmer-v1 task. Unfortunately, we observed that the proposed P3S-TD3 algorithm could not solve the deceptive gradient problem in the Swimmer-v1 task either. Fig. 9 shows the learning curves of P3S-TD3 and TD3 algorithm. In Khadka & Tumer (2018), the authors proposed an effective evolutionary algorithm named ERL to solve the deceptive gradient problem on the Swimmer-v1 task, yielding the good performance on Swimmer-v1, as shown in Fig. 9. P3S-TD3 falls short of the performance of ERL on Swimmer-v1. However, it is known that CEM-TD3 discussed in Appendix D outperforms ERL on other tasks (Pourchot & Sigaud (2019)). Furthermore, we observed that P3S-TD3 outperforms CEM-TD3 on most environments in Appendix D.

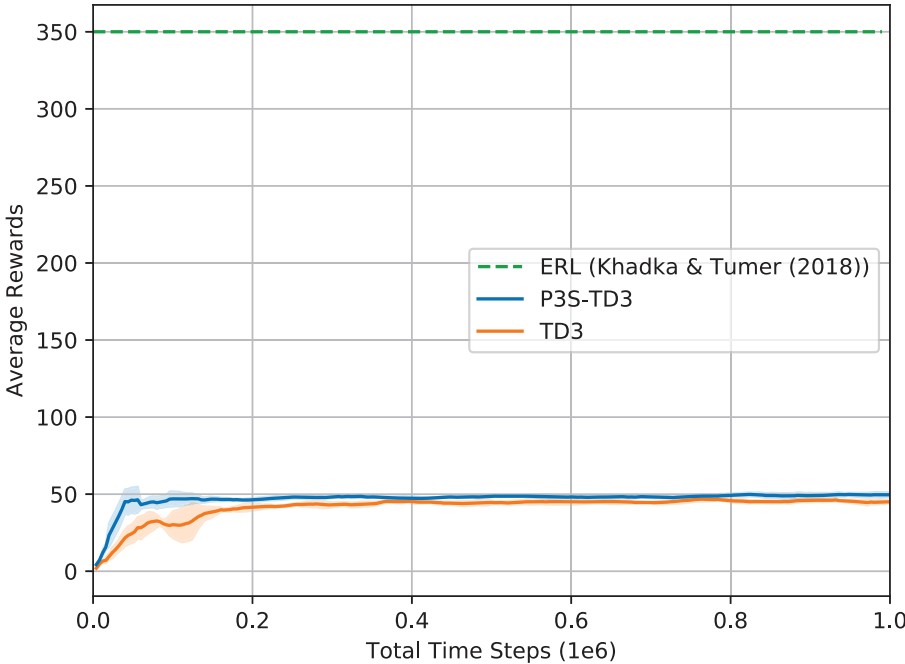

Figure 9: Performance on Swimmer-v1 of P3S-TD3 (blue), TD3 (orange), and the final performance of evolutionary RL (Khadka & Tumer (2018), green dashed line).

## APPENDIX G. THE TWIN DELAYED DEEP DETERMINISTIC POLICY GRADIENT (TD3) ALGORITHM

The TD3 algorithm is a current state-of-the-art off-policy algorithm and is a variant of the deep deterministic policy gradient (DDPG) algorithm (Lillicrap et al. (2015)). The TD3 algorithm tries to resolve two problems in typical actor-critic algorithms: 1) overestimation bias and 2) high variance in the approximation of the Q-function. In order to reduce the bias, the TD3 considers two Q-functions and uses the minimum of the two Q-function values to compute the target value, while in order to reduce the variance in the gradient, the policy is updated less frequently than the Q-functions. Specifically, let $Q_{\theta_1}$, $Q_{\theta_2}$ and $\pi_\phi$ be two current Q-functions and the current deterministic policy, respectively, and let $Q_{\theta_1'}$, $Q_{\theta_2'}$ and $\pi_{\phi'}$ be the target networks of $Q_{\theta_1}$, $Q_{\theta_2}$ and $\pi_\phi$, respectively. The target networks are initialized by the same networks as the current networks. At time step $t$, the TD3 algorithm takes an action $a_t$ with exploration noise $\epsilon$: $a_t = \pi_\phi(s_t) + \epsilon$, where $\epsilon$ is zero-mean Gaussian noise with variance $\sigma^2$, i.e., $\epsilon \sim \mathcal{N}(0, \sigma^2)$. Then, the environment returns reward $r_t$ and the state is switched to $s_{t+1}$. The TD3 algorithm stores the experience $(s_t, a_t, r_t, s_{t+1})$ at the experience replay buffer $\mathcal{D}$. After storing the experience, the Q-function parameters $\theta_1$ and $\theta_2$ are updated by gradient descent of the following loss functions:

$$L(\theta_j) = \hat{\mathbb{E}}_{(s,a,r,s')\sim\mathcal{D}} \left[(y - Q_{\theta_j}(s,a))^2\right], \quad j = 1, 2 \tag{79}$$

where $\hat{\mathbb{E}}_{(s,a,r,s')\sim\mathcal{D}}$ denotes the sample expectation with an uniform random mini-batch of size $B$ drawn from the replay buffer $\mathcal{D}$, and the target value $y$ is given by

$$y = r + \gamma \min_{j=1,2} Q_{\theta_j'}(s', \pi_{\phi'}(s') + \epsilon), \quad \epsilon \sim \text{clip}(\mathcal{N}(0, \tilde{\sigma}^2), -c, c). \tag{80}$$

Here, for the computation of the target value, the minimum of the two target Q-functions is used to reduce the bias. The procedure of action taking and gradient descent for $\theta_1$ and $\theta_2$ are repeated for $d$ times ($d = 2$), and then the policy and target networks are updated. The policy parameter $\phi$ is updated by gradient descent by minimizing the loss function for $\phi$:

$$L(\phi) = -\hat{\mathbb{E}}_{s\sim\mathcal{D}} \left[Q_{\theta_1}(s, \pi_\phi(s))\right], \tag{81}$$

and the target network parameters $\theta_j'$ and $\phi'$ are updated as

$$\theta_j' \leftarrow (1 - \tau)\theta_j' + \tau\theta_j \qquad \phi' \leftarrow (1 - \tau)\phi' + \tau\phi. \tag{82}$$

The networks are trained until the number of time steps reaches a predefined maximum.

APPENDIX H. PSEUDOCODE OF THE P3S-TD3 ALGORITHM

---

**Algorithm 1** The Population-Guided Parallel Policy Search TD3 (P3S-TD3) Algorithm

---

**Require:** $N$: number of learners, $T_{initial}$: initial exploration time steps, $T$: maximum time steps, $M$ : the best-policy update period, $B$: size of mini-batch, $d$: update interval for policy and target networks.
1: Initialize $\phi^1 = \cdots = \phi^N = \phi^b$, $\theta_j^1 = \cdots = \theta_j^N$, $j = 1, 2$, randomly.
2: Initialize $\beta = 1, t = 0$
3: **while** $t < T$ **do**
4:    $t \leftarrow t + 1$ (one time step)
5:    **for** $i = 1, 2, \cdots, N$ in parallel **do**
6:       **if** $t < T_{initial}$ **then**
7:          Take a uniform random action $a_t^i$ to environment copy $\mathcal{E}^i$
8:       **else**
9:          Take an action $a_t^i = \pi^i\left(s_t^i\right) + \epsilon$, $\epsilon \sim \mathcal{N}(0, \sigma^2)$ to environment copy $\mathcal{E}^i$
10:      **end if**
11:      Store experience $(s_t^i, a_t^i, r_t^i, s_{t+1}^i)$ to the shared common experience replay $\mathcal{D}$
12:   **end for**
13:   **if** $t < T_{initial}$ **then**
14:      **continue** (i.e., go to the beginning of the while loop)
15:   **end if**
16:   **for** $i = 1, 2, \cdots, N$ in parallel **do**
17:      Sample a mini-batch $\mathcal{B} = \{(s_{t_l}, a_{t_l}, r_{t_l}, s_{t_l+1})\}_{l=1,\ldots,B}$ from $\mathcal{D}$
18:      Update $\theta_j^i$, $j = 1, 2$, by gradient descent for minimizing $\tilde{L}(\theta_j^i)$ in (83) with $\mathcal{B}$
19:      **if** $t \equiv 0(\mathrm{mod}\ d)$ **then**
20:         Update $\phi^i$ by gradient descent for minimizing $\tilde{L}(\phi^i)$ in (84) with $\mathcal{B}$
21:         Update the target networks: $(\theta_j^i)' \leftarrow (1-\tau)(\theta_j^i)' + \tau\theta_j^i$, $(\phi^i)' \leftarrow (1-\tau)(\phi^i)' + \tau\phi^i$
22:      **end if**
23:   **end for**
24:   **if** $t \equiv 0(\mathrm{mod}\ M)$ **then**
25:      Select the best learner $b$
26:      Adapt $\beta$
27:   **end if**
28: **end while**

---

In P3S-TD3, the $i$-th learner has its own parameters $\theta_1^i$, $\theta_2^i$, and $\phi^i$ for its two Q-functions and policy. Furthermore, it has $(\theta_1^i)'$, $(\theta_2^i)'$, and $(\phi^i)'$ which are the parameters of the corresponding target networks. For the distance measure between two policies, we use the mean square difference, given by $D(\pi(s), \tilde{\pi}(s)) = \frac{1}{2}\|\pi(s) - \tilde{\pi}(s)\|_2^2$. For the $i$-th learner, as in TD3, the parameters $\theta_j^i$, $j = 1, 2$ are updated every time step by minimizing

$$\tilde{L}(\theta_j^i) = \hat{\mathbb{E}}_{(s,a,r,s')\sim\mathcal{D}}\left[(y - Q_{\theta_j^i}(s, a))^2\right] \tag{83}$$

where $y = r + \gamma \min_{j=1,2} Q_{(\theta_j^i)'}(s', \pi_{(\phi^i)'}(s') + \epsilon)$, $\epsilon \sim \mathrm{clip}(\mathcal{N}(0, \tilde{\sigma}^2), -c, c)$. The parameter $\phi^i$ is updated every $d$ time steps by minimizing the following augmented loss function:

$$\tilde{L}(\phi^i) = \hat{\mathbb{E}}_{s\sim\mathcal{D}}\left[-Q_{\theta_1^i}(s, \pi_{\phi^i}(s)) + \mathbf{1}_{\{i\neq b\}}\frac{\beta}{2}\left\|\pi_{\phi^i}(s) - \pi_{\phi^b}(s)\right\|_2^2\right]. \tag{84}$$

For the first $T_{initial}$ time steps for initial exploration we use a random policy and do not update all policies over the initial exploration period. With these loss functions, the reference policy, and the initial exploration policy, all procedure is the same as the general P3S procedure described in Section 3. The pseudocode of the P3S-TD3 algorithm is shown above.

## APPENDIX I. HYPER-PARAMETERS

**TD3** The networks for two Q-functions and the policy have 2 hidden layers. The first and second layers have sizes 400 and 300, respectively. The non-linearity function of the hidden layers is ReLU, and the activation functions of the last layers of the Q-functions and the policy are linear and hyperbolic tangent, respectively. We used the Adam optimizer with learning rate $10^{-3}$, discount factor $\gamma = 0.99$, target smoothing factor $\tau = 5 \times 10^{-3}$, the period $d = 2$ for updating the policy. The experience replay buffer size is $10^6$, and the mini-batch size $B$ is 100. The standard deviation for exploration noise $\sigma$ and target noise $\tilde{\sigma}$ are 0.1 and 0.2, respectively, and the noise clipping factor $c$ is 0.5.

**P3S-TD3** In addition to the hyper-parameters for TD3, we used $N = 4$ learners, the period $M = 250$ of updating the best policy and $\beta$, the number of recent episodes $E_r = 10$ for determining the best learner $b$. The parameter $d_{min}$ was chosen among $\{0.02, 0.05\}$ for each environment, and the chosen parameter was 0.02 (Walker2d-v1, Ant-v1, Delayed Hopper-v1, Delayed Walker2d-v1, Delayed HalfCheetah-v1), and 0.05 (Hopper-v1, HalfCheetah-v1, Delayed Ant-v1). The parameter $\rho$ for the exploration range was 2 for all environments. The time steps for initial exploration $T_{initial}$ was set as 250 for Hopper-v1 and Walker2d-v1 and as 2500 for HalfCheetah-v1 and Ant-v1.

**Re-TD3** The period $M'$ was chosen among $\{2000, 5000, 10000\}$ (MuJoCo environments) and $\{10000, 20000, 50000\}$ (Delayed MuJoCo environments) by tuning for each environment. The chosen period $M'$ was 2000 (Ant-v1), 5000 (Hopper-v1, Walker2d-v1, HalfCheetah-v1), 10000 (Delayed HalfCheetah-v1, Delayed Ant-v1), and 20000 (Delayed Hopper-v1, Delayed Walker2d-v1).

