# OpenReview forum: "Population-Guided Parallel Policy Search for Reinforcement Learning"
_ICLR.cc/2020/Conference — Accept (Poster)_

### Official Review · AnonReviewer2 · 2019-10-08
**Official Blind Review #2**

**Rating:** 3

**Review:**

This work presents a distributed framework for off-policy RL consisting of multiple agents that are trained in parallel while also being regularized to be similar to the current best policy. Maintaining a population of agents can help mitigate issues due to convergences to local optima. The method is evaluated on standard continuous control tasks, and shows some performance improvements over methods that train just a single agent. Ablation experiments are also conducted to evaluate the effects of different design decisions.

The overall method is sensible and performance looks promising. But the technical innovation is fairly modest. As the authors pointed out, the proximal constraint has been used in previous distributed RL framework, and the main difference in this work is enforcing a trust region penalty against the best policy, as opposed to an average policy. The proof of guaranteed improvement largely follows the proof from other trust region methods, by replacing the old policy with the best policy from the previous iteration. The experiments did compare with a number of previous algorithms, but the comparisons are all to algorithms that train a single agent at a time, and no comparisons are made to other distributed RL algorithms. Including some comparisons to other distributed methods can help strengthen the claims in favour P3S. In particular, how does regularizing using the best policy compare to using an average policy like in Distral [The et al., 2017]? That being said, the performance improvements on most environments appear to be fairly modest, and it is not clear if the improvements are indeed significant. Including additional experiments on more challenging tasks could be helpful here. Overall, the contribution of this work is pretty incremental, and I think it is not quite at the standards for ICLR at this time. But with more thorough evaluation and polishing, I think this work can make for a strong submission.

More specific notes:

There are a fair bit of awkward phrasing and grammatical errors in the writing, which hurts the clarity of the exposition.

One of the advantages of a distributed framework is faster wall-clock time. The current learning curves only compare the sample count. It might also be informative to compare the wall-clock time of the different methods, as well as including comparisons to other distributed frameworks.

Some experiments to show how performance scales with the number of learners can also be helpful. Since one of the possible factors that improve performance for P3S is having multiple agents, providing some insight on how performance varies with different numbers of agents can be valuable in this regard.


**Experience Assessment:**

I have published one or two papers in this area.

**Review Assessment: Checking Correctness Of Derivations And Theory:**

I assessed the sensibility of the derivations and theory.

**Review Assessment: Checking Correctness Of Experiments:**

I carefully checked the experiments.

**Review Assessment: Thoroughness In Paper Reading:**

I read the paper thoroughly.

---

> ### Author Response · Authors · 2019-11-15
> **Response to Reviewer 2**
>
> [Comment]: Level of innovation
>
> [Response]: We agree with the reviewer on that the innovation of the proposed P3S scheme is modest in some sense. However, we also think that there exists definitely some new ingredients in the proposed P3S scheme. During the revision, we compared the P3S scheme with more other parallel learning schemes including latest algorithms based on evolutionary approaches and center policy like in Distral/Divide-and-Conquer. It is now observed that the prosed P3S scheme outperforms these up-to-date algorithms. This shows the effectiveness of the proposed P3S method.
> Furthermore, we provide theoretical guarantee for monotone average performance improvement for P3S, while many algorithms just proposed algorithms only. As indicated by the reviewer, certain portion of the proof is from other papers, but the proof has non-trivial and non-straightforward steps to derive the desired result and the required condition for monotone improvement, which is expressed as Assumption 2 and implemented practically based on the proposed beta update rule.  We believe this is some definite contribution.
>
>  [Comment]: Comparing other distributed RL methods and a method using a center policy (average policy)
>
> [Response]: We could not compare the performance with PBT for finding hyperparameters and suggested other variants of the resetting method because of the time limit. Instead, we were able to compare P3S-TD3 with CEM-TD3 (Pourchot & Sigaud (2019)) which is a state-of-the-art evolutionary RL algorithm, and Center-TD3 which is based on Distral (Teh et al. (2017))/Divide-and-Conquer (Ghosh et al. (2018)). It is observed that the proposed P3S-TD3 algorithm outperforms CEM-TD3 and algorithm based on Distral/Divide-and-Conquer on all the considered MuJoCo and  delayed MuJoCo environments (Hopper-v1, Walker2d-v1, HalfCheetah-v1, Ant-v1 and their delayed versions) except delayed HalfCheetah-v1. Thus, this shows the effectiveness of the proposed P3S method. Please see Appendices D and E in the revised paper.

---

### Official Review · AnonReviewer1 · 2019-10-09
**Official Blind Review #1**

**Rating:** 8

**Review:**

The  paper proposes a new approach to multi-actor RL, which ensure diversity and performance of the population of actors, by distilling the policy of the best performing agent in a soft way and maintaining some distance between the agents. The authors show improved performance over several state-of-the-art mono-actor algorithms and over several other multi-actor RL algorithms.

I'm in favor of accepting the paper despite a few serious weaknesses described below.

A good point of the paper is the related work section which provides a good and concise survey of various multi-actor RL approaches: distributed RL, population-based training and Guided Policy Search (the last part about exploiting best information looks less relevant).

Here a set of random  remarks:

About  population-based training, there is quite a lot of repetition between the introduction and the related work  section.

The P3S approach described in the beginning of Section 3 and the end of Section 3.2 seems to rely on some arbitrary choices and a few hyperparameters whose impact is not much studied.

In the theoretical study (Section 3.1), a KL divergence is used as a distance between policies, which implies stochastic policies. But P3S is used on top of TD3, where policies are deterministic. This should definitely be discussed.

The role of \beta in the theory is not put forward in a way to make the point of Section 3.2 clear. The theory should be clarified in this respect.

The top of p5 is made of a unique sentence over 7 lines which is completely obfuscating. This part must be rewritten and much clarified.

I would be glad to see the performance on Swimmer, as this benchmark is known to suffer from  a deceiptive gradient.

The fact that a negative cost of action in Ant-v1 can result in no action in this environment is reminiscent of the same effect shown in the simpler Continuous Mountain Car environment in the Gep-PG paper (Colas, Oudeyer and Sigaud, ICML 2018). Actually, moving to simpler benchmarks would make it possible to provide more detailed empirical studies of the inner mechanisms of the P3S approach.

 p6: "The policies used for evaluation are stochastic for PPO and ACKTR, and deterministic for the others." Do you mean you used a deterministic policy for SAC? This would be unusual, as SAC with a deterministic policy is very close to TD3.

 p6: "In Fig.   3,  it is first observed that the performance of TD3 here is similar to that in the originalTD3 paper (Fujimoto et al. (2018)), and the performance of other baseline algorithms is also similar to that in the original papers (Schulman et al. (2017); Haarnoja et al. (2018))". This sentence can be much compressed: "In Fig.   3,  it is observed that all other baseline algorithms is also similar to that in the original papers (Fujimoto et al. (2018),Schulman et al. (2017); Haarnoja et al. (2018))".

Why did you use the v1 versions of the benchmarks, and not the v2?

I did not check the proofs in appendix.

typos:

p2: hyperparamters
p7: is the way how the best =>  is the way the best...
p7: other all parallel => all other parallel
p22: environmet

**Experience Assessment:**

I have read many papers in this area.

**Review Assessment: Checking Correctness Of Derivations And Theory:**

I did not assess the derivations or theory.

**Review Assessment: Checking Correctness Of Experiments:**

I assessed the sensibility of the experiments.

**Review Assessment: Thoroughness In Paper Reading:**

I read the paper at least twice and used my best judgement in assessing the paper.

---

> ### Author Response · Authors · 2019-11-15
> **Response to Reviewer 1**
>
> [Comment]: Rewriting words and sentences for more clarity
>
> [Response]: During the revision, we rewrote many parts of the paper for readability and clarity.
>
> [Comment]: Sensitivity on hyperparameter setting
>
> [Response]: The major hyperparameters of P3S is the best policy update interval M, rho and d_min. As mentioned in Section 4, we used M=250 and rho=2 for all the considered environments whether they are the original MuJoCo or delayed version. Furthermore, we also used d_min= 0.02 or 0.05 for all the considered environments in Section 4 and Appendices. So, it seems that we do not need to fine-tune the hyperparameters for each environment.
>
> [Comment]: Application of the KL divergence to deterministic policies
>
> [Response]: Note that the KL divergence between two Gaussian distributions with (mu_1, sigma_1) and (mu_2, sigma_2) is
>              log(sigma_2 / sigma_1) + (1/2) [sigma_1^2 + (mu_1 – mu_2)^2] / [ sigma_2^2] - 1/2.
> A deterministic policy is expressed as a function a(s), where s is the input state and a(s) is the output action. For two deterministic policies a_1(s) and a_2(s), we can construct two stochastic Gaussian policies \pi_{i, Gaussian}(a| s), i=1,2 with the same standard deviation sigma=sigma_1=sigma_2,  mu_1=a_1(s), and mu_2 = a_2(s).  As sigma decreases, the Gaussian policy converges to the corresponding deterministic policy. For the two Gaussian policies with the same standard deviation sigma, the KL divergence above reduces to
>             (mu_1 – mu_2)^2 / sigma^2 = (a_1(s) – a_2(s))^2/ sigma^2.
> Hence, the KL divergence is proportional to the mean square error of two actions.
>
> [Comment]: Adaptation of \beta
>
> [Response]: Please see the response to common comments.
>
> [Comment]: Performance on Swimmer and Continuous Mountain Car
>
> [Response]: The authors in (Khadka & Tumer (2018), Pourchot & Sigaud (2019)) noticed that most deep RL methods suffer from a deceptive gradient problem on the Swimmer-v1 task, and most RL methods could not learn effectively on the Swimmer-v1 task. Unfortunately, we observed that the proposed P3S-TD3 algorithm could not solve the deceptive gradient problem in the Swimmer-v1 task either.  Please see Appendix F.
> Due to the limited time for revision, we could not simulate the task of continuous mountain car.
>
> [Comment]: Stochastic or deterministic policy in SAC for evaluation
>
> [Response]: There seems a confusion. We used stochastic policy to train the policy in SAC. However, as described in the original SAC paper (Haarnoja et al. (2018)), the performance of the policy using only the mean (i.e., deterministic version) has better performance than stochastic policy.  Hence, the original SAC paper uses stochastic policy for training and deterministic policy for evaluation. Therefore, to give more favor to SAC, we also followed the training and evaluation procedure described in the original SAC paper.
>
> [Comment]: About the v1 version of tasks.
>
> [Response]: V1 tasks are the same as V2 tasks in the aspects of the shape of robots, observation space, action space, and reward function. Only the internal MuJoCo file version is different. V1 is based on MuJoCo file 1.31 and V2 is based on MuJoCo file 1.50. Therefore, the performance difference between the two versions is less. Most of the considered baseline algorithms used V1. Hence, for consistency and reproducibility, we used V1.

---

### Official Review · AnonReviewer3 · 2019-10-19
**Official Blind Review #3**

**Rating:** 6

**Review:**

The authors propose another method of doing population-based training of RL policies. During the training process, there are N workers running in N copies of the environment, each with different parameter settings for the policies and value networks. Each worker pushes data to a shared replay buffer of experience. The paper claims that a natural approach is to have a chief job periodically poll for the best worker, then replace the weights of each worker with the best one. Whenever this occurs, this reduces the diversity within the population.

In its place, the authors propose a soft-update in the chief. At every merging cycle, the chief queries which worker performs best. If that worker is worker B, it emits pi_B's parameters to each of the other workers. Instead of replacing the parameters exactly, worker i's loss is then augmented by beta * D(pi_i, pi_B), where D is some distance measure that is measured over states sampled from the replay buffer. The "soft" update encourages individual workers to match pi_B without directly replacing their parameters, which maintains diversity in the population. In this work, pi is always represented by a deterministic policy and D is the mean-squared-error in action space (this is argued as equivalent to the KL divergence between the two policies if the policies were represented by Gaussian with the same, constant standard deviation). The beta parameter is updated online using heuristics based on how D(pi_i, pi_B) compares to D(pi_i, old_pi_i). Using TD3 as a base algorithm, the population-based version performs better, and there are ablations for various parts of the population algorithm.

I thought this paper was interesting, but thought it was strange that there were very few comparisons to other population / ensemble-based training methods. In particular they mention the copying problem as a downside of population-based training (PBT), but do not compare against PBT at all. Additionally, my understanding of PBT is that when they replace bad agents with the best agent, they only replace the worst performing agents (not all of them), and they additionally add some random perturbations to their hyperparameter settings. This goes counter to the claim that they collapse the population to a single point- by my reading the exploration step avoids this collapse.

An experiment I'd like to see is trying PBT, where different workers do in fact use different hyperparameters. My understanding is that in P3S-TD3 there is a single hyperparameter setting shared across all workers (plus some hyperparameters deciding the soft update).

I'd also like to see ablations for the Resetting variant (Re-TD3), where only the bottom half or 2/3rds of the workers are reset. This would give empirical evidence for the "population collapse" intuition - we should expect to see some improvements if we avoid totally collapsing the population, while still copying enough to partially exploit the current best setting.

Many inequalities in the paper are argued by compare the expectation of negative Q of one policy to the negative Q of another - I believe the derivations would be much easier to follow if the authors simply multiplied all sides by -1 and adjusted inequalities accordingly. It is much easier to think about Q-value-1 > Q-value-2 rather than -Q-value-1 < -Q-value-2 when trying to interpret what the equation is saying.

For related work, papers on evolutionary strategies and the various self-play-in-a-population papers seem relevant, since these often take the form of having each worker i do a different perturbation that is later merged by a chief.

In Figure 4 it feels weird that results are the regular Mujoco envs for 2 problems and the delayed envs for the other 2 problems. When looking at the appendix, it's rather clearly cherry picked to show the best results in favor of PS3-TD3. I would prefer the Delayed MuJoCo experiments be in a separate figure, or to include the TD3/SAC/ACKTR/PPO/etc. results for the delayed envs as well (these don't appear to be in the appendix)

On the theoretical results: the 1st assumption seems very strong. The first assumption argues that pi_B is always 1-step better than pi_old for every state. That assumption already takes you very far towards arguing "updating pi_old to pi_B is good". The 2nd assumption is more reasonable but I'm confused how rho and d play into the theoretical results. Do they play any role in how much the policy is expected to improve, or do the constants just need to exist?

The last comment on the theory side is that I still don't understand the intuition for why we want to learn beta such that

KL(pi_new || pi_b) = max {rho * KL_max(pi_new || pi_old), d}

In the practical algorithm, beta is updated online to increase / decrease the importance of the "match pi_B" term if the ratio between the two strays too far from 1 (with the threshold set to [1/1.5, 1.5] in a manner similar to PPO's approach). But why should it be important for the two values to be close to one another? Let me write out the derivation continuing from Eqn (57) in the appendix.

With a substitution that doesn't use (c) to drop the beta * (KL - KL) term, we get

E_{pi_b}[-Q_new] >= E_{pi_new}[-Q_new] + beta * (KL - KL)
-->
E_{pi_new}[Q_new] >= E_{pi_b}[Q_new] + beta * (KL - KL)

Then, in Theorem 1, we recursively apply this inequality, accumulating a number of beta * (KL - KL) terms. In the end we get

Q_new >= (discounted sum rewards from pi_b) + (discounted sum of beta * (KL - KL) with expectation over states from pi_b)
= Q_pi_b + (sum of beta *(KL - KL) terms)

By my reading, shouldn't this mean we want KL(pi_new || pi_b) - max {rho * KL_max(pi_new || pi_old), d} to be as large as possible, rather than 0? The more positive this term is, the more improvement we get between Q_new and Q_pi_b.

--------------------------

Overall, this feels like a good paper, but I'm not too familiar with prior empirical results for population-based RL methods. The ablations suggested that pretty much any reasonable population-based method outperformed using a single worker, and because of this it seems especially important to have ablations to other population-based prior work, rather than just variants of its own method.

I would be okay with this paper as-is despite some of its flaws, but think it could be better pending rebuttal.

Edit: I have read the author reply and other reviews. I do not plan to change my rating but do think the paper is improved by the added baselines and better explanation of what the beta adaptation rule is doing. I would ask the authors to make sure this description is as clear as possible, as the argued improvement gap seems central to the work.

**Experience Assessment:**

I have published one or two papers in this area.

**Review Assessment: Checking Correctness Of Derivations And Theory:**

I assessed the sensibility of the derivations and theory.

**Review Assessment: Checking Correctness Of Experiments:**

I carefully checked the experiments.

**Review Assessment: Thoroughness In Paper Reading:**

I read the paper at least twice and used my best judgement in assessing the paper.

---

> ### Author Response · Authors · 2019-11-15
> **Response to Reviewer 3**
>
> [Comment]: On Theorem 1, performance improvement gap, and the adaptation rule for \beta
>
> [Response]: Please see the response to common comments.
>
> [Comment]: Comparison with other methods such as PBT, other variants of the resetting method, and evolutionary algorithm.
>
> [Response]: We could not compare the performance with PBT for finding hyperparameters and suggested other variants of the resetting method because of the time limit. Instead, we were able to compare P3S-TD3 with CEM-TD3 (Pourchot & Sigaud (2019)) which is a state-of-the-art evolutionary RL algorithm, and Center-TD3 which is based on Distral (Teh et al. (2017))/Divide-and-Conquer (Ghosh et al. (2018)). It is observed that the proposed P3S-TD3 algorithm outperforms CEM-TD3 and algorithm based on Distral/Divide-and-Conquer on all the considered MuJoCo and delayed MuJoCo environments (Hopper-v1, Walker2d-v1, HalfCheetah-v1, Ant-v1 and their delayed versions) except delayed HalfCheetah-v1. Thus, this proves the effectiveness of the proposed P3S method. Please see Appendices D and E in the revised paper.
>
> [Comment]: Results of the single-agent baselines on delayed MuJoCo environments
>
> [Response]: In the revised paper, we have included the performance of the single-agent baselines on delayed MuJoCo environments in Appendix C. It is observed that P3S-TD3 outperforms these single-agent baselines except the case of ACKTR in delayed Ant-v1.

---

### Author Response · Authors · 2019-11-15
**Common Response**

We thank all reviewers for their valuable comments. During the rebuttal period, we revised the paper to incorporate the reviewers’ comments. We hope that the revised paper and this response satisfy all the reviewers’ concerns. The modified part in the revised paper is in blue.

[Comment]: On the connection to the theoretical result Theorem 1 and the adaptation method for \beta

[Response]: As noticed by Reviewer 3, there exists a performance improvement gap between the updated policy in the current update period and the best policy in the previous update period in Theorem 1. In the revised paper, we considered this gap explicitly, and Theorem 1 and its proof were modified accordingly. Now, the meaning of the proposed adaptation method of \beta is clear. The proposed adaptation method of \beta intends to maximize the improvement gap. We rewrote the beta update part in Section 3.2 and have included Appendix B to explain the beta update rule. Please see Section 3.2 and Appendix B for the beta adaptation rule and Appendix A for the modified proof of Theorem 1.

[Comment]: More results on comparison with other population-based or parallel algorithms

[Response]: Reviewers 2 and 3 suggested more results on comparison with other population-based algorithms or parallel algorithms. In Section 4.3 of the first submission, we considered several population-based methods such as distributed RL, the experience-sharing only method, the resetting method, where the resetting method resembles PBT (Jaderberg et al. (2017)). We realized that obtaining distributed performance results such as wall clock time on a real distributed simulation environment is beyond the scope of an academic institute. Instead, following the comments on more comparison results, during the revision we considered and implemented several up-to-date parallel learning algorithms such as CEM-TD3 (Pourchot & Sigaud (2019)) and Distral (Teh et al. (2017))/Divide-and-Conquer (Ghosh et al. (2018)). Here, CEM-TD3 was chosen as the latest evolutionary algorithm and Distral (Teh et al. (2017))/Divide-and-Conquer (Ghosh et al. (2018)) was chosen as algorithms based on a center policy. It is observed that the proposed P3S-TD3 algorithm outperforms CEM-TD3 and algorithm based on Distral/Divide-and-Conquer on all the considered MuJoCo and  delayed MuJoCo environments (Hopper-v1, Walker2d-v1, HalfCheetah-v1, Ant-v1 and their delayed versions) except delayed HalfCheetah-v1. Thus, this shows the effectiveness of the proposed P3S method. Please see Appendices D and E in the revised paper.

[Comment]: Rewriting words and sentences for more clarity

[Response]: During the revision, we rewrote many parts of the paper for readability and clarity.

---

### Decision · Program_Chairs · 2019-12-19

**Decision:**

Accept (Poster)

**Comment:**

The  paper proposes a new approach to multi-actor RL, which ensure diversity and performance of the population of actors, by distilling the policy of the best performing agent in a soft way and maintaining some distance between the agents. The authors show improved performance over several state-of-the-art mono-actor algorithms and over several other multi-actor RL algorithms.  Initially, reviewers were concerned with magnitude of the contribution/novelty, as well as some technical issues (e.g. the beta update), and relative lack of baseline comparisons.  However, after discussion the reviewers largely agree that their main concerns have been addressed.  Therefore, I recommend this paper for acceptance.